# Relying on the Metrics of Evaluated Agents

## Abstract

Online platforms and regulators face a continuing problem of designing effective evaluation metrics. While tools for collecting and processing data continue to progress, this has not addressed the problem of *unknown unknowns*, or fundamental informational limitations on part of the evaluator. To guide the choice of metrics in the face of this informational problem, we turn to the evaluated agents themselves, who may have more information about how to measure their own outcomes. We model this interaction as an agency game, where we ask: *When does an agent have an incentive to reveal the observability of a metric to their evaluator?* We show that an agent will prefer to reveal metrics that differentiate the most difficult tasks from the rest, and conceal metrics that differentiate the easiest. We further show that the agent can prefer to reveal a metric *garbled* with noise over both fully concealing and fully revealing. This indicates an economic value to privacy that yields Pareto improvement for both the agent and evaluator. We demonstrate these findings on data from online rideshare platforms.

## CCS Concepts

• **Information systems** → **Incentive schemes**; • **Security and privacy** → Economics of security and privacy.

## Keywords

information elicitation, evaluation metrics, principal-agent games, unknown unknowns

**ACM Reference Format:**
Anonymous Author(s). 2018. Relying on the Metrics of Evaluated Agents. In *Proceedings of Make sure to enter the correct conference title from your rights confirmation emai (Conference acronym 'XX)*. ACM, New York, NY, USA, 29 pages. https://doi.org/XXXXXXX.XXXXXXX

## 1 Introduction

> *"Only the wearer knows where the shoe pinches."* –Proverb

The design of effective evaluation metrics continues to be of central concern to online platforms and their surrounding economies, including regulators, engineers, and users. Indeed, the problem has only grown in importance in recent years, as advances in the ability of online platforms to collect and analyze data have led to a reliance on data-driven evaluation metrics to steer decision-making. The design problem becomes an adaptive and incentive-theoretic one.

For instance, in the rapidly evolving AI marketplace, government regulators are increasingly concerned with the evaluation of AI systems, focusing attention on the evaluation of safety, performance, and liability, in the context of evolving law and policy [27]. Private companies have demonstrated investment in improving evaluation as well, from developing leaderboards [5, 21] to funding evaluation efforts from third-party researchers [8]. Academic researchers are at the forefront of efforts to develop new evaluation metrics that target an ever-increasing spectrum of capabilities [23, 36].

Within the firms developing these technologies, evaluation has also long been a core part of operations, where internal evaluation metrics drive company decisions in all areas, from engineering to advertising to pricing. Online platforms (like recommender systems, digital marketplaces, and gig economy platforms) draw from rich data sources to compute their internal evaluation metrics. Still, significant challenges remain in the design and usage of these metrics, from handling the limitations and biases of current "proxy" metrics [9, 47], to identifying which aspects of the data are most relevant to a given decision [15, 42].

We consider the development of evaluation metrics, where *we use the word "metric" to loosely refer to a variable that an evaluator uses to infer the value or difficulty of a task under consideration.* For example, a score on an evaluation benchmark is a metric that indicates the value of a LLM trained by a firm. The delivery time is a metric that indicates both value and difficulty of a food delivery completed by a driver. The h-index is a metric that some employers may use to infer the productivity of a researcher. Metrics almost never tell the whole story, and vary in informativeness, but are nonetheless deeply embedded in online platforms and economies, driving the evolution of technology and shaping its societal impact.

For both firms and regulators, the modern prevailing challenge with the design of metrics is not necessarily with the *ability* to collect large volumes of data, but of discerning which aspects of the data are relevant to the task at hand. This knowledge in turn informs allocation of resources for data-collection efforts, and downstream decisions that affect users like pricing and auditing. We consider evaluators who are powerful, but with a limited worldview: they can verify and collect any data for metrics that they are aware of, but their worldview is limited in that they are not aware of all possible metrics that could be important.

A core perspective underlying this work is that while evaluators might be limited by their worldviews, this does not mean that better information does not exist in the wider socio-technical system. In fact, those who know the most about a task are often none other than the agents who perform it themselves. These evaluated agents might both know and be willing to share better evaluation metrics. Thus, in this work, we study the *incentives* for information transfer from an evaluated agent to the evaluator.

More generally, there is a long history of eliciting information and feedback from the wider socio-technical system for the purposes of evaluation and design. For example, the subfield of *participatory design* in human-computer interaction focuses on methods

for designing technical systems with user input [45]. For AI evaluation, some of the biggest companies currently developing the AI systems are also involved in the evaluation process, even cited as direct collaborators in a recent White House briefing [27]. In online content and labor platforms, there is often a rich reliance on feedback channels like surveys and focus groups of creators and workers on the platform.

We present *a theoretical model of incentives for an agent to share metrics* across such channels. We characterize the incentives that influence an evaluated agent to share metrics with an evaluator, with a particular focus on the unique capabilities and limitations of modern online platforms. Specifically, we consider a setting in which there are significant asymmetries in data collection, data verification, and price-setting power between the evaluator and the evaluated agents. Yet, a key information asymmetry remains in the other direction, where the evaluated agents hold unique information about what features are most correlated with their success. A core strategic dependence in our model is the relationship between the agent and the metrics—we consider the incentives an agent may experience with respect to the revelation of metrics that impact their own evaluation by an outside entity.

The incomplete nature of metrics has been discussed in the seminal work of Holmström and Milgrom [26] on contracts with multidimensional tasks, where a principal may observe only a subset of dimensions that are relevant to their value or the agent's cost. When the principal is aware of which dimensions are missing, Holmström and Milgrom [26] show how to optimally reward the observed dimensions given properties of the agent's cost structures. The distinguishing feature of our setting is that we consider the unobserved dimensions to be *unknown unknowns*, yielding a model that is in a sense inherently non-Bayesian. We focus on an information-transfer mechanism where the agent has the power to possibly reveal these hidden dimensions to the principal.

Specifically, we examine an agent's incentives for information sharing through the lens of an *agency game with information transfer*. We build on a classical agency game where a principal contracts an agent to complete a task, and the principal only has partial information about the agent's costs[1] when setting a contract. To capture the agent's additional information and opportunity to improve the metrics by which they are evaluated, our model supposes that the agent is privately aware of additional variables that correlate with their cost of task completion, and further has the opportunity to reveal these additional variables to the principal prior to the design of the contract. We analyze when the agent prefers a contract that depends on the revealed metrics over a contract that ignores them.

Importantly, our model deviates from the standard Bayesian setup in persuasion games, since without the agent's help, the principal is not even aware of the *existence* of the metric. We also assume the agent cannot lie about or selectively mask the metric depending on its value, reflecting a common reality where online platforms and regulators often have significant data collection power, and can even compel an evaluated agent to report metrics [18].

While the principal is always better off knowing more metrics, the incentives for the agent are nuanced—revealing a metric reduces

the amount of information rent the agent can extract when their costs turn out to be low. However, better information also means more high-cost tasks will go forward that the agent otherwise would not have accepted, as the optimal contract would adequately reward the agent when the metric indicates a high cost.

We prove that the agent prefers to reveal metrics that strongly differentiate the highest cost settings from rest, and conceal metrics that strongly differentiate the lowest cost settings from the rest. For example, a university might want to reveal to a government funder the number of low-income students that matriculate (indicating higher operating costs), but conceal the amount of funding they anticipate from private donations (indicating that they could continue without additional funds).

Our analysis becomes richer when we expand the agent's action space to include the ability to *garble* or add noise to their metric before revealing it. We analyze a garbling mechanism that guarantees a notion of local differential privacy for the metric, and show that under a fairly wide set of conditions, the agent may prefer to reveal a garbled metric over both fully concealing and fully revealing the original metric. In fact, garbling can lead to *Pareto improvement* for both the principal and agent. From a policy standpoint, this demonstrates settings when both a principal and agent can derive economic value from a privacy preserving mechanism, even before considering the inherent social value of privacy.

Finally, we demonstrate an application of our theory to analyzing feature discovery for pricing on rideshare platforms, using public data scraped from Uber and Lyft.

*Our contributions can be summarized as follows:*

(1) We introduce a model for the elicitation of unknown metrics, via an agency game with information transfer.
(2) We present sufficient conditions under which an agent would prefer to reveal or conceal a metric.
(3) We show that when the agent has the ability to reveal a garbled metric, this can lead to Pareto improvement for both the principal and agent.
(4) We leverage connections between our model and price discrimination to analyze total welfare.
(5) We apply our model on real data to analyze feature discovery on for pricing on rideshare platforms.

## 2 Related Work

Our model builds on literature from contract design and agency games. It also fits into a large literature on information design, and can be seen as an instance of information design with a specific structure. Our model also overlaps with a large body of work on price discrimination.

*Agency games and contract design.* We build from the well established contract design problem of Laffont and Tirole [34], which concerns a principal's design of a contract when an agent's effort and cost type are privately held by the agent. Key to this setting are asymmetries in values and information between the principal and agent, and the literature explores issues of moral hazard and adverse selection that arise from these asymmetries [24, 33, 40].

A fundamental result regarding signaling incentives in contract theory is Holmström [25]'s sufficient statistic theorem, which showed that it benefits an agent for the contract to be conditioned

---

[1]In addition to representing the difficulty of the task, the agent's "cost" could also be interpreted as the external market value for the task, or the price that the principal has to beat for the agent to complete the task for them.

on any information that is independently informative of the agent's effort. Our setting models an agent's incentive to share information about its cost type, which yields different results from the analysis of effort signaling, and is closer to some analyses of persuasion games which we discuss in more detail below. Milgrom [39] also classifies the "favorableness" of signals to an agent, presenting monotonicity properties that we also leverage in this work.

*Persuasion games and information design.* The use of stakeholder-supplied information for decision-making was introduced by Milgrom and Roberts [38] in their seminal work on *persuasion games*. Milgrom and Roberts [38] give an example of analyzing a buyer's purchasing strategy when a seller can send a quality signal about their product. While this broad motivation of information transfer from interested parties is close to our setting of eliciting metrics from evaluated agents, our model has a key distinction that without the agent's help, the principal could not even access a prior over the metric. We also constrain the agent's information revelation strategy to not be able to depend on the realized value of the variable. This brings our setting closer to a problem of *metric discovery*, where the fundamental problem is a principal's lack of awareness of a metric's observability, rather than a realized signal value [46].

More broadly, Bayesian persuasion and information design provides a general framework for analyzing the effects of the distribution of information on the outcomes of a game [13, 30]. Our model can be seen as a specific instantiation of an information design problem where the *sender* is the agent, the *receiver* is the principal, and the contracting relationship determines the principal's and agent's action spaces and equilibria. We impose a constraint over the sender's information transfer policy in order to capture the incentives for an agent to reveal *observability* of a variable to the principal. Notably, the agent cannot selectively mask or lie about their signal based on its realized value, as our model is motivated by settings where a principal has powerful data collection capabilities.

*Price discrimination.* Our model has analogies with classical price discrimination, and thereby opens new avenues for applying the well-developed tools from price discrimination to understanding the metric design problem. It also uncovers new challenges that extend existing price discrimination perspectives. These issues have been anticipated by Bergemann et al. [12], who define a mapping from third-degree price discrimination onto the class of agency problems and establish the existence of market segmentations that achieve all possible trade-offs between consumer and producer surplus within some basic constraints. Our work complements this analysis—instead of analyzing all possible segmentations, we consider the agent's information-revelation incentives as a function of the properties of a specific segmentation induced by some cost-correlated variable which is initially only available to the agent.

Our garbling setting also notably differs from the models of price discrimination with transportation costs and arbitrage [52] or restricted price discrimination [4]. While restricted price discrimination mechanisms also effectively interpolate between full discrimination and no discrimination, we prove a substantive difference between garbling and these mechanisms in Appendix G.5.

*Sunspots and correlated equilibria.* Our model is also connected to the so-called sunspots literature [28, 51]. In this literature, there are multiple steady-state equilibria and an observable random variable produces a correlated equilibrium, with agents conditioning their behavior on this otherwise extraneous variable not because it matters to payoffs, but because it predicts others' behaviors. The literature is known as sunspots because, during the 19th century, some people believed sunspot activity predicted agricultural yields [29], and while it did not, it could predict the behavior of commodity traders who believed it did. This literature is the polar opposite of the problem we study: sunspots are a known, extraneous variable believed to be relevant, while we study an unknown (to the principal) relevant variable.

*Preferences for privacy.* Our model interacts with literatures on privacy and information-hiding, by highlighting situations where agents experience conflicting incentives both for and against sharing information. The result can be that agents may prefer partial sharing, which echoes the subtleties that arise elsewhere when privacy and behavior interact [2, 17].

Several works have considered the interaction between privacy and price discrimination in particular [3, 7, 16, 20, 22, 41]. Close to our setting is Fallah et al. [20], who consider a noise addition mechanism that distorts the principal's view of the observed value distributions per market segment. However, unlike our model, they assume that the principal does not know the aggregate market (the marginal distribution of costs in our setting), and their noise model can distort the principal's belief about the aggregate market. Our garbling mechanism also differs from these other works in various ways (including, e.g., that there is no cost to noise addition, that the agent does not derive inherent value from garbling, and that noise is only added to the metric). Still, these various models have led to similar findings that privacy can yield economic benefits.

## 3  Agency Game with Information Transfer

To gain insight into the incentives surrounding the discovery of metrics, we start with a standard agency game in which a principal contracts an agent to complete a task. In such a game, we ask, *when does the agent have an incentive to reveal observability of a cost-correlated variable to the principal?*

Specifically, suppose a principal contracts the agent to complete a task, where an agent may exert binary effort. Suppose the principal receives value $b$ if the agent exerts effort and completes the task, and 0 otherwise (we assume that the task is completed deterministically if the agent exerts effort). Suppose the agent incurs cost $C \in \mathbb{R}_+$ for exerting effort. The exact cost is unobserved to the principal, but the principal is aware of a prior distribution over agent's cost type, denoted by the random variable $C$. The agent observes both the proposed price and their realized cost type before deciding whether or not to exert effort. This maps onto the well-established agency game setup where the principal must design a contract when the agent's cost and effort are private [34].

To model the agent's additional knowledge of a metric, suppose the agent is aware of a variable $X \in \mathcal{X}$ which is correlated with their cost $C$. However, $X$ is an *unknown unknown* to the principal. Prior to the principal's design of the contract, the agent has a choice of whether or not to *reveal* $X$, which entails revealing both the *observability* of $X$ at the time of the design of the contract, and the realized *value* of $X$ at the execution of the contract (which the principal can verify). The fact that $X$ is not observable to the principal without the agent revealing is a key distinction between

our model and prior work in signaling games. That is, the principal cannot rely on prior information over $C$ and $X$, since they are not even aware of the *existence* of $X$. This allows us to analyze a decision problem that differs from that of the standard Bayesian framework, which instead focuses on an agent's decision to reveal their *value* of $X$ after it is realized.

If the agent chooses to not reveal $X$, then the rest of the game proceeds as a standard agency game with private cost: the principal designs a contract based on their knowledge of the prior distribution $C$. If the agent reveals $X$, then the principal can offer a contract that conditions on the realized value of $X$. The timing of the full agency game with the possibility of agent information transfer is summarized in Figure 1. The text in black matches the standard agency game [34], and the text in gray represents additional elements introduced by our model.

The principal's contract design problem when $X$ is concealed reduces to choosing a single price $p$ where the agent is paid $p$ if the task is completed, and zero otherwise. If $X$ is revealed, then the principal offers a contract with distinct prices $\rho(x)$ for different realized values of $x \in X$. At the execution of the contract, the agent receives payment $\rho(x)$ if the task is completed and $X = x$, and 0 otherwise. We assume that the principal still receives the same value $b$ if the task is completed, regardless of $X$.

The key question in this work concerns the agent's decision of whether or not to reveal $X$ at time $t = 1$. In Section 4, we treat this as a binary decision of whether or not to reveal $X$; we will later relax this in Section 5 to expand the agent's action space to reveal a garbled version of $X$, thus interpolating between the concealed and revealed settings.

## 3.1 Optimal Contracts

We begin by outlining the optimal contract and equilibrium utilities of the principal and agent when the contract is agnostic of $X$, which we refer to as the *concealed information* setting. Then, we outline the optimal contract and equilibrium utilities when the principal can condition on $X$ to determine payments to the agent, which we refer to as the *revealed information* setting. We assume that both principal and agent are risk neutral throughout.

*Concealed information contract.* If the metric $X$ is not revealed at $t = 1$, then the rest of the game proceeds as a standard agency game with private cost, where the principal chooses a single transfer $p$ based on their knowledge of the prior distribution over the agent's cost $C$. The agent's optimal policy at the execution of the contract is to exert effort if their realized cost is less than or equal to the payment. Thus, the agent's best response to the principal's choice of transfer $p$ is to exert effort with probability $F(p) = \mathbb{P}(C \leq p)$ (the set where $C = p$ has measure zero).[2]

For a given choice of transfer $p$, the principal's expected utility when $X$ is hidden is given by $\Pi_{\text{con}}(p) := F(p)(b - p)$. The agent's expected utility under the principal's choice of transfer $p$ is given by $V_{\text{con}}(p) := \mathbb{E}[(p - C) \mathbb{1}(C < p)]$. The principal moves first and chooses $p^* \in \arg\max_{p \geq 0} \Pi_{\text{con}}(p)$. The agent's utility at equilibrium is then $V_{\text{con}}(p^*)$.

---

[2]We may also think of $F(p)$ as the task completion "quantity" as a function of price $p$, or the proportion of agents drawn uniformly at random from a population with costs distributed as $C$ that would complete the task for price $p$.

*Revealed information contract.* If the metric $X$ is revealed at time $t = 1$, then the principal can vary the payment amount depending on $X$, denoted as $\rho : X \to \mathbb{R}_+$. The agent's best response to the principal's payment function $\rho(\cdot)$ is to exert effort with probability $F_x(\rho(x))$ when $X = x$. The principal's expected utility in the revealed setting is given by $\Pi_{\text{rev}}(\rho) := \mathbb{E}[F_X(\rho(X))(b - \rho(X))]$, maximized at $\rho^* \in \arg\max_{\rho \in \mathcal{F}} \Pi_{\text{rev}}(\rho)$. The agent's expected utility under the principal's choice of transfer function $\rho(\cdot)$ is then $V_{\text{rev}}(\rho) := \mathbb{E}[(\rho(X) - C) \mathbb{1}(C < \rho(X))]$. The agent's choice of whether to reveal or conceal is ultimately determined by their *utility difference*, $V_{\text{rev}}(\rho^*) - V_{\text{con}}(p^*)$.

## 4 Welfare Effects of Information Revelation

Our first central goal is to analyze the circumstances under which the agent would prefer to either conceal or reveal the observability of the metric $X$ at time $t = 1$. We show that the agent prefers to conceal $X$ when it strongly differentiates the lowest cost setting from the rest, and reveal $X$ when it strongly differentiates the highest cost setting from the rest. We also analyze the consequences of the resulting decision on total welfare, connecting our model to the literature on the effects of price discrimination on total welfare.

### 4.1 Agent's Revelation Incentives

To understand the properties of a feature $X$ that would lead the agent to either want to reveal or conceal, we begin with a family of *thresholding features*, $\{X^t : t \in \mathbb{R}^+\}$, where $X^t = 0$ if $C > t$, and $X^t = 1$ if $C \leq t$. That is, $X^t = 1$ indicates a low cost type, and $X^t = 0$ indicates a low cost type, thresholded by $t$. Our main results are two theorems that indicate that if $t$ is small enough, the agent will prefer to conceal, and if $t$ is large enough (relative to the principal's value $v$), then the agent will prefer to reveal. The main intuitive takeaway is that *the agent will prefer to conceal features that strongly identify a low cost type, and will prefer to reveal features that strongly identify a high cost type.*

We now present precise conditions in which the agent prefers to reveal or conceal a thresholding feature $X^t$. We begin with two regularity conditions on the cost distribution $C$.

**Assumption 1** (Differentiable cost distribution). *The cost $C$ has density $f$ and CDF $F(c)$ which is concave for $c \geq p^*$.*

The concavity assumption on $F$ connects the optimal price $p^*$ to first-order conditions on the principal's utility.

**Assumption 2** (Monotone reverse hazard rate). *The ratio $\frac{F(c)}{f(c)}$ is strictly monotone increasing for $c > 0$.*

This assumption reflects the log-concavity of $F$, and mirrors the standard monotone hazard rate condition [10, 11, 37].

Under these assumptions, we show that the agent prefers to reveal $X^t$ for sufficiently high $t$, as long as the principal's task value $b$ is not too low and not too high.

THEOREM 1 (AGENT PREFERS TO REVEAL FOR HIGH THRESHOLDS). *Under Assumptions 1 and 2, if the cost is bounded above by $\bar{C}$ with $f(\bar{C}) > 0$, and if $b \in \left(\bar{C}, \bar{C} + \frac{1}{f(\bar{C})}\right)$, then there exists a threshold $\underline{t}$ such that for all $t > \underline{t}$, the agent prefers to reveal $X^t$.*

| P and A share prior over $C$. A knows prior joint distribution of $C, X$. | A decides whether to reveal observability of $X$, including joint distribution of $C, X$. | P offers a contract, which can depend on $X$ if revealed. | $X$ is realized. A learns their cost type $C$. | A decides whether or not to accept the contract. | The contract is executed and utilities realized. |
|---|---|---|---|---|---|
| $t = 0$ | $t = 1$ | $t = 2$ | $t = 3$ | $t = 4$ | $t = 5$ |

**Figure 1: Timing of the agency game with information transfer between principal (P) and agent (A).**

To gain intuition for why the principal's task value $b$ matters, consider an extremely high task value of $b \gg \bar{C}$. In this case, the principal's optimal hidden price will already be $p^* = \bar{C}$, and the agent can only end up with lower prices from revealing information. Thus, we need that $b$ is at least low enough that the optimal price $p^* < \bar{C}$. On the other hand, if $b$ is too low, the principal will not price high enough to accommodate the highest cost agents, even after $X^t$ is revealed. Theorem 1 shows that for $b$ in a "sweet spot," the agent will prefer to reveal a thresholding feature $X^t$ that differentiates the highest cost agents from the rest of the crowd. We give a more precise characterization of the exact threshold $\underline{t}$ as a function of $b$ in Theorem 3 in the Appendix.

On the opposite end of the spectrum, we next show that the agent will prefer to *conceal* any $X^t$ which differentiates the *lowest* cost agents from the rest. To illustrate this, let $\Delta(t)$ be the difference in the agent's value between revealing and concealing $X^t$: $\Delta(t) :=$ $V_{\mathrm{rev}}(p^*) - V_{\mathrm{con}}(p^*)$ for $X = X^t$. The agent prefers to reveal $X^t$ if and only if $\Delta(t)$ is positive. Note that $\Delta(t)$ must be zero at the endpoints since $X^t$ would be entirely uninformative about $C$ (See Lemma 6 in the Appendix).

We next show that $\Delta(t)$ initially decreases in $t$, meaning that the agent would prefer to conceal $X^t$ for such sufficiently small $t$. To do this, we require one more assumption on the agent's sensitivity to price.

**Assumption 3** (Price elasticity). *The agent's price elasticity for task completion is high at the optimal hidden price:* $\eta(p^*) \geq 1$, *where* $\eta(p) = \frac{pf(p)}{F(p)}$.

The price elasticity $\eta(p)$ measures the sensitivity of the agent's task completion quantity to the price offered by the principal. A price elasticity of at least 1 indicates that a percentage change in price affects the quantity at least as much. Under this assumption, we now formalize the agent's incentive to conceal for low $t$.

**THEOREM 2** (AGENT PREFERS TO CONCEAL FOR LOW THRESHOLDS). *Under Assumptions 1, 2, and 3, if $f(0) > 0$, then $\Delta'(0) < 0$.*

Theorem 2 completes the picture for $t$ sufficiently low. As $t$ increases marginally from zero, the agent's value for revelation decreases below zero, such that the agent prefers concealment.

In summary, the analysis of the thresholding features $X^t$ gives a picture of the types of features that the agent would prefer to reveal or conceal: if the feature $X$ differentiates the highest cost agents from the crowd, the agent prefers to reveal. If the feature $X$ differentiates the lowest cost agents from the crowd, the agent prefers to conceal. For an illustration of these theorems using the uniform distribution, see Figure 5 in the Appendix. In Section 6, we show experiments with more realistic features from a rideshare dataset that that reflect our theoretical findings here.

We give more general conditions the agent to prefer to reveal or conceal for $X$ beyond thresholding features in Appendix D.2. These more general conditions are less easy to interpret, but nonetheless verifiable for any given feature distribution.

## 4.2 Total Welfare Consequences

We now consider whether revealing $X$ increases total welfare, or the sum of utilities of the principal and agent. Note that the principal always benefits from revelation, which we formalize in Appendix D.4. When $X$ is concealed, total welfare is given by $W_{\mathrm{con}}(p) :=$ $V_{\mathrm{con}}(p) + \Pi_{\mathrm{con}}(p)$, and when $X$ is revealed, total welfare is given by $W_{\mathrm{rev}}(\rho) := V_{\mathrm{rev}}(\rho) + \Pi_{\mathrm{rev}}(\rho)$.

The question of whether price discrimination increases total welfare has been well studied [48]. Our comparison of the concealed vs. revealed contracts has a direct isomorphism with third-degree monopoly price discrimination. Specifically, our concealed setting corresponds to monopoly pricing without price discrimination, with the seller acting as principal and the buyer acting as agent. Our revealed setting corresponds to a monopoly seller enacting third-degree price discrimination over markets segmented by $X$.

Thus, with minor adjustments, we can apply results from the price discrimination literature that characterize the effects of third-degree monopoly price discrimination on total welfare. Mirroring Varian [48]'s seminal work, Lemma 1 shows that total welfare increases only if the quantity of tasks completed also increases in the revealed setting compared to the concealed setting.

**Lemma 1.** *Total welfare increases under revelation (i.e.* $W_{\mathrm{rev}}(\rho^*) > W_{\mathrm{con}}(p^*)$*) only if task completion quantity increases:* $F(p^*) < \mathbb{E}[\mathbb{1}(C < \rho^*(X)]$.

Task completion quantity does not always increase under revelation, and it is thus possible for total welfare to decrease under revelation—we give an example of this in Appendix E.

## 5 Information Revelation with Garbling

So far, we have compared the settings when $X$ is either concealed or revealed, focusing on the agent's incentive to induce each setting at time $t = 1$. We now generalize the agent's action space to instead be able to reveal a *garbled* version of the variable $X$. The agent's garbling action space interpolates between the concealed and revealed settings. We consider a *randomized response* garbling mechanism, which is a noise addition method that has been applied in many settings, from statistical informativeness [14] to survey experiment design [50] to differential privacy [19, 32]. In particular, we formally show that our garbling mechanism guarantees a notion of differential privacy with respect to an agent's metric value $X$.

As an overview of results in this section, we show that there exist conditions under which the agent would prefer to garble over

both concealment and revelation. Thus, having the option to garble can benefit the agent, and even induce the agent to reveal more information when they would otherwise fully opt to conceal $X$, thus leading to a Pareto improvement for both the principal and agent. We also show that garbling improves total welfare over the concealed setting.

## 5.1 Agency Game with Garbling

Suppose the agent has the option to present the principal with a different variable $Y$ with noise added to the binary variable $X$:

$$Y = \begin{cases} X & \text{w.p. } \varepsilon \\ \xi & \text{w.p. } 1 - \varepsilon, \end{cases}$$

where $\xi \sim \text{Bernoulli}(\theta)$ is independent of $X$ and $C$, for $\theta = \mathbb{P}(X = 1)$. For ease of exposition, we focus on binary $X$ and $Y$, though further extensions with different noise models can be made for continuous $X$.

The game with garbling proceeds as before, but the agent selects $\varepsilon \in [0, 1]$ at time $t = 1$. The full timing is outlined in Figure 2. The agency game with garbled information transfer is a generalization of the previous game: in the previous timing in Figure 1, the agent's choice at $t = 1$ would be equivalent to selecting $\varepsilon$ from a more restricted set $\{0, 1\}$. In other words, garbling *interpolates* between the concealed and revealed settings in the previous game.

The principal treats the variable $Y$ as a revealed metric with prior joint distribution $Y, C$, and proceeds to design the optimal contract conditioning on $Y$. This is optimal for the principal both in the case where the principal has no knowledge of $\varepsilon$ or $X$, and in the case where the principal knows $\varepsilon$ and the joint distribution of $C, X$ (but not the realized value of $X$).

The game proceeds as delineated for the revealed information contract in Section 3.1, but using the variable $Y \in \mathcal{Y}$ instead of $X$. Specifically, the principal designs a contract with prices $\rho : \mathcal{Y} \to \mathbb{R}_+$, and upon execution of the contract, the agent receives payment $\rho(y)$ if they exert effort and $Y = y$.

The principal's utility under revelation of a given garbled variable $Y$ is $\Pi_{\text{garb}}(\rho) \coloneqq E[(b - \rho(Y)) \mathbb{1}(C < \rho(Y))]$, and the agent's utility is $V_{\text{garb}}(\rho) \coloneqq E[(\rho(Y) - C) \mathbb{1}(C < \rho(Y))]$. After $Y$ is revealed, the principal selects $\rho^* \in \arg\max_{\rho \in \mathcal{F}} \Pi_{\text{garb}}(\rho)$.

For a given $\varepsilon$ chosen at $t = 1$, we denote the equilibrium utilities as $\Pi_{\text{garb}}(\varepsilon) = \Pi_{\text{garb}}(\rho^*), V_{\text{garb}}(\varepsilon) = V_{\text{garb}}(\rho^*)$. The agent's optimal choice of $\varepsilon$ at time $t = 1$ is then $\varepsilon^* \in \arg\max_{\varepsilon \in [0,1]} V_{\text{garb}}(\varepsilon)$. An optimal choice of $\varepsilon^* = 0$ corresponds to full concealment, and $\varepsilon^* = 1$ corresponds to full revelation. In the following sections, we analyze the effects of this optimal choice on welfare, and compare this to the welfare effects of the agent's optimal choice in the previous game without the garbling option. We also formally connect the garbling model to existing notions of differential privacy.

## 5.2 Agent's Garbling Incentives

Our primary observation is that even for distributions satisfying the regularity conditions from before, there exist cases when some amount of garbling is preferred over both concealment and revelation, for *both* orderings of concealment vs. revelation. That is, there exist settings when $V_{\text{garb}}(\varepsilon^*) > V_{\text{rev}}(\rho^*) > V_{\text{con}}(p^*)$ (a.k.a. garbled

> revealed > concealed), and settings when $V_{\text{garb}}(\varepsilon^*) > V_{\text{con}}(p^*) > V_{\text{rev}}(\rho^*)$ (garbled > concealed > revealed).

Both orderings are significant from a policy standpoint. The case when garbled > revealed > concealed is interesting since it indicates that the agent derives economic value from adding noise to their data compared to fully revealing. Thus, *even without any inherent value for privacy, the agent still prefers the privatized setting.*

The case when garbled > concealed > revealed is perhaps even more interesting, since without the ability to only partially reveal a garbled $Y$, the agent would have otherwise chosen to conceal. Thus, allowing for garbling results in a strict Pareto improvement for both the principal and agent; that is, *both principal and agent benefit from the option of privacy.* This provides an incentive for a platform to agree to maintain garbling of a feature, even if they could otherwise discover $X$ after $Y$ is revealed.

Figure 3 illustrates a continuum of distributions that contain both of these orderings. Specifically, let $C$ be a mixture of exponentials with $C|X = 0 \sim \text{Exp}(\frac{1}{\lambda_0})$ and $C|X = 1 \sim \text{Exp}(\frac{1}{\lambda_1})$, where $\lambda_x$ is the mean of the distribution. Fixing $\theta = \frac{1}{2}$, $b = 1$, and $\lambda_0 = 0.5$, Figure 3 shows a range of $\lambda_1$ in which the agent prefers garbled > concealed > revealed, and as $\lambda_1$ increases, this flips to garbled > revealed > concealed. Thus, even within the same family of "nice" distributions, both orderings can occur.

To give a more general theoretical characterization of the agent's garbling incentives, we give sufficient conditions for the agent to prefer a nonzero amount of garbling over full revelation in Appendix G.2. This can be seen as a softer version of the previous analysis of when the agent prefers full concealment over full revelation.

## 5.3 Garbling and Total Welfare

We next show that garbling increases total welfare ($W_{\text{garb}}(\varepsilon) \coloneqq \Pi_{\text{garb}}(\varepsilon) + V_{\text{garb}}(\varepsilon)$) relative to concealment. The principal always benefits from more information, which we formalize in G.4. Optimal garbling always increases total welfare over concealment. A more subtle finding is that adding a marginal amount of garbling compared to concealment *also* increases total welfare.

First, we show that the *optimal* amount of garbling chosen by the agent also increases total welfare over the fully concealed setting.

**Lemma 2** (Optimal garbling increases welfare over concealment). Let $\varepsilon^* \in \arg\max_{\varepsilon \in [0,1]} V_{\text{garb}}(\varepsilon)$. Then $W_{\text{garb}}(\varepsilon^*) \geq W_{\text{garb}}(0)$.

Intuitively, Lemma 2 follows from the fact that the principal is never hurt by additional information (formalized in Appendix G.4). Given that the agent benefits from their optimal garbling choice, total welfare must increase.

Next, we show that relative to the fully concealed setting, the marginal effect of revealing any information on total welfare is initially positive relative to full concealment.

**Lemma 3** (More information initially increases welfare). Increasing $\varepsilon$ from 0 marginally increases total welfare: $W'_{\text{garb}}(0) \geq 0$. The inequality is strict if $\Pi'_{\text{garb}}(0) > 0$.

While the agent's chosen amount of garbling $\varepsilon^*$ improves total welfare over concealment, the question remains of where $\varepsilon^*$ falls relative to the optimal amount of garbling that maximizes total welfare. Lemma 4 shows that the optimal amount of garbling that

| P and A share prior over $C$. | A selects $\varepsilon$, and reveals | P offers a | $X$ and $Y$ are | A decides whether | The contract is |
|---|---|---|---|---|---|
| A knows prior joint distribution of $C, X$. | observability of $Y$, including joint distribution of $C, Y$. | contract, which depends on $Y$. | realized. A learns their cost type $C$. | or not to accept the contract. | executed and utilities realized. |
| $t = 0$ | $t = 1$ | $t = 2$ | $t = 3$ | $t = 4$ | $t = 5$ |

**Figure 2: Timing of the agency game with *garbled* information transfer between principal (P) and agent (A).**

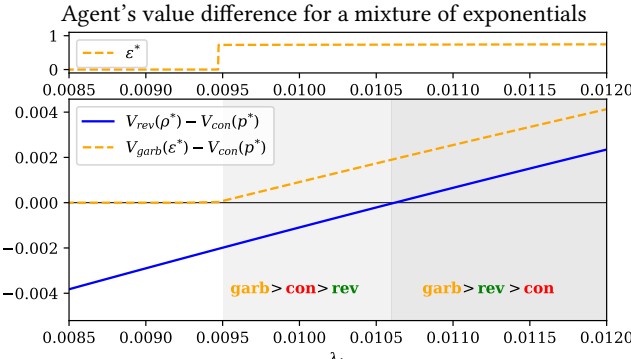

Figure 3: Agent's utility differences for revealing $X$ and garbled $Y$ when $C$ is a mixture of exponentials. We fix $\theta = \frac{1}{2}$, $b = 1$, and $\lambda_0 = 0.5$, and vary $\lambda_1$. The **solid blue line** shows the agents utility difference upon revealing $X$ for each value of $\lambda_1$. The **dashed orange line** shows the agent's utility difference upon revealing the optimal garbled $Y$, for optimal garbling parameter $\varepsilon^*$ displayed above. This first lighter shaded region highlights settings where **garbled** > **concealed** > **revealed**. The second darker shaded region highlights settings where **garbled** > **revealed** > **concealed** .

maximizes total welfare must necessarily reveal at least as much information as the optimal amount of garbling chosen by the agent.

**Lemma 4** (More information increases total welfare relative to agent optimal garbling). $W'_{\text{garb}}(\varepsilon) \geq V'_{\text{garb}}(\varepsilon)$ for all $\varepsilon \in [0, 1]$. The inequality is strict if $\Pi'_{\text{garb}}(0) > 0$.

The benefits of garbling to total welfare have interesting policy implications, as the increase in total welfare means that a third party designer could inject garbling noise (perhaps for privacy reasons), and later redistribute surplus such that both principal and agent do not end up with worse utilities. We next formalize the connection between garbling and notions of differential privacy.

### 5.4 Garbling and Differential Privacy

As we model it, garbling directly induces a guarantee of local differential privacy for the agent's feature value $X$. Upon garbling, the principal can still observe the marginal distribution of the feature $X$, but they have limited ability to identify any specific agent's value of $X$. We formalize this below.

While randomized response and other differentially private mechanisms are typically thought of as methods to preserve the confidentiality of sensitive information motivated by the inherent value of privacy, our model focuses on the possible economic benefits

to both principal and agent from such a mechanism. Our main theorem shows that both the agent and the principal can derive utility from the agent garbling $X$, even *without* accounting for any inherent value for privacy. In this section, we formally relate our garbling mechanism to the literature on differential privacy.

Whenever an agent's metric value $X$ is queried, it first passes through the garbling mechanism. This acts as a local randomizer Any adversary viewing the output of a local randomizer would have limited ability to identify the agent's original $X$ value.

**Definition 1** (Local Randomizer [32]). A mechanism $\mathcal{M} : \mathcal{X} \to \mathcal{Y}$ is an $\varepsilon$-local randomizer if for all $x, x' \in \mathcal{X}$ and all $y \in \mathcal{Y}$, $\mathbb{P}(\mathcal{M}(x) = y) \leq e^{\varepsilon}\mathbb{P}(\mathcal{M}(x') = y)$.

Any mechanism that only accesses the underlying data by way of local randomizers is known to enjoy a guarantee of *local differential privacy* [32] with a privacy parameter that reflects the composition of the local randomizers' $\varepsilon$ parameters across multiple data accesses.

Our garbling mechanism is a local randomizer, and therefore any prices that are based on $X$ solely through the garbling mechanism are locally differentially private.

**Lemma 5** (Privacy of garbling mechanism). For $\theta \in (0, 1)$, for all $\varepsilon \leq O(\theta)$, the garbling mechanism is an $O(\varepsilon)$-local randomizer with respect to $X$. A pricing algorithm that accesses $X$ only by way of the garbling mechanism is $O(\varepsilon)$-locally differentially private.

## 6 Experiments

To demonstrate how our theory can apply to real scenarios and feature distributions, we turn to public data from the rideshare platforms Uber and Lyft. Platforms like these employ algorithmic pricing methods that depend on numerous ride metrics (or features). We investigate an agent's incentives to reveal features to the platform that would affect their pricing in several hypothetical scenarios. Our findings track with the intuition built by our theory, and illustrate a variety of agent dynamics that can occur.

*Dataset.* We consider the Uber and Lyft data that is publicly available on Kaggle [43]. This data was scraped using API queries over the course of one week at the end of November, 2018 in Boston, MA. The dataset includes a set of Uber rides and a set of Lyft rides, and contains columns for price, distance, surge multiplier, and others variables. Implementation details are given in Appendix I.

*Scenario: revelation of a highly cost-correlated feature.* Consider a simple scenario in which a competing rideshare company (principal) wants to win over drivers (agents) from Lyft. We use the price column as an approximation for the price offered to the driver. The cost variable $C$ is the price that Lyft offers for the ride, since this is the price that the principal must beat for the driver to switch to the principal's platform. The agent gains utility if the principal offers price $p > C$ to induce the agent to switch; otherwise, the

agent ignores the offer and is no worse off than before. Our central question is whether the agent would be better off if the principal offers prices as a function of $X$ = distance, which is highly correlated with $C$. In other words, *would the agent prefer to reveal a highly cost-correlated feature to the initially naive principal?*[3]

*Results.* We analyze the agent's utilities for revealing, concealing, and garbling, and compare with the predictions from our theory. We present results when the principal's value for the agent's switch is $b = 1.5\bar{C}$, such that the prices are not degenerate (e.g., always offering the maximum cost or never offering above some amount).[4]

First, the agent indeed prefers to reveal the distance feature $X$, with the exact difference in the agent's value between revelation and concealment shown in Figure 4. This is particularly interesting because a feature which *exactly* matches the cost $C$ would leave the agent with a surplus of 0. This means that the distance feature is correlated enough with $C$ to change the principal's prices, but adds enough noise to still leave the agent with some surplus. The Pearson correlation between $X$ and $C$ is $\approx 0.36$.

We now compare this to the agent's incentive to reveal a coarser, binarized version of the distance feature. Let $Z^t = 1$ if $X \leq t$ and 0 otherwise. This binarized version tracks closely with our theory on thresholding features, as $Z^t$ can be thought of as a noisy thresholding of $C$. Figure 4 shows that for some $t$, the agent actually prefers to reveal $Z^t$ over the full feature $X$. Figure 4 also shows that the agent prefers to conceal $Z^t$ for $t$ very low, and reveal for $t$ is very high, which matches Theorems 1 and 2.

If the agent is further given the chance to *garble* $Z^t$, we show that sometimes the agent prefers garbling over revelation. Figure 4 shows the agent's optimal garbling amount $\varepsilon^*$ for each $Z^t$. For the highest and lowest thresholds, the agent sticks to either full revelation or full concealment; but for some $t$ in between, the agent achieves even higher utility through garbling than full revelation. As a practical implication, this constitutes a set of scenarios in which an agent derives economic value from feature privacy.

*Discussion and Limitations.* The findings from this scenario give a rough picture of the types of features that an agent would want to reveal given this data. In Appendix I.1, we construct a second similar scenario in which the principal starts with a more sophisticated baseline pricing model, which yields similar results. In general, these are not intended as a characterization of real rideshare platforms, but instead as a demonstration of how our model can be applied to analyze metric elicitation incentives in practical settings. For example, a collective action organization acting on behalf of drivers might perform analysis like the above, and determine whether it would be beneficial to reveal new, possibly garbled or thresholded features to a rideshare platform. We encourage replication of similar analyses by agencies with internal data sources.

## 7 Conclusions and Future Work

We have presented a model to analyze the discovery of metrics in a setting of information asymmetry where relevant metrics are

---

[3]Most real rideshare companies are likely aware of the importance of distance, but for the purposes of this illustration, we consider distance as a stand-in for similarly highly cost-correlated variables.

[4]A high value of $b$ can be thought of as the principal receiving high reward for winning users to their platform, which is often important to investors. We repeat this for different values of $b$ in the Appendix.

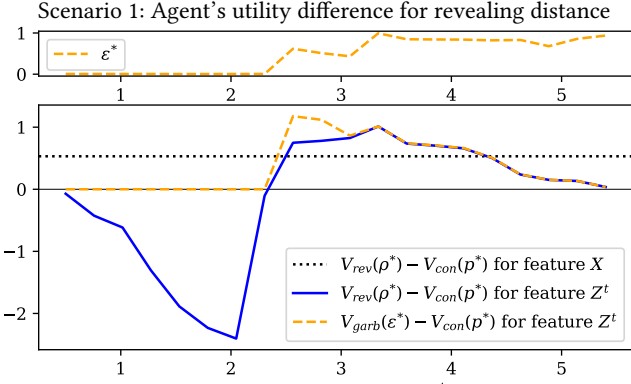

Scenario 1: Agent's utility difference for revealing distance

**Figure 4: Agent's utility differences for revealing the distance feature (positive means the agent prefers to reveal). The dotted line is the agent's value difference upon revealing the full distance feature $X$. The solid blue line shows the agents value differences upon revealing $Z^t$ for different thresholds $t$. The dashed orange line shows the agent's value difference upon revealing the optimal garbled version of $Z^t$, for optimal garbling parameter $\varepsilon^*$ displayed above ($\varepsilon = 1$ corresponds to full revelation, and $\varepsilon = 0$ to full concealment). In alignment with our theory, the agent prefers to conceal for $t$ low enough, and reveal for $t$ high enough. There also exist $t$ values in the middle in which garbled > revealed > concealed.**

unknown to a principal, but known to an evaluated agent. In characterizing the conditions under which new metrics are revealed, we have shown that an evaluated agent will prefer to reveal metrics that differentiate their highest cost settings from the rest, and conceal metrics that differentiate their lowest cost settings from the rest. Furthermore, the agent may actually prefer to reveal a *garbled* version of the metric over both fully concealing and fully revealing, which demonstrates settings in which both agent and principal may still derive economic value from the option of privacy, even without adding their inherent value for privacy.

More broadly, this work was motivated by analyzing a mechanism by which one might discover *unknown unknowns*. Even as data and computational methods become increasingly sophisticated and widely available, this problem of discovery of *which* metrics or variables to analyze continues to permeate the natural sciences, social sciences, and engineering. In machine learning contexts in particular, there has been growing recent work on markets for sharing data [1, 6, 31], but the question of incentives for sharing *features* is distinct, and can be analyzed using the model presented here.

Ultimately, there are many other possibilities for for formulating the question of *who* holds relevant information, and *when* they would be willing to share it. For example, one could model incentives for third-party individuals to offer new metrics, or perhaps bi-directional information transfer where a principal and agent both hold distinct information. The key element of our work that may be worth retaining in alternative information design frameworks is the property that the variable itself may be unknown to the information receiver.

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

## A  Additional Notation

In the following proofs, we apply the following additional notation.

We define $F_x(c) := \mathbb{P}(C \le c | X = x)$ as the CDF of the conditional distribution of $C$ given $X$. Let $f_x(c)$ denote the corresponding density.

Let $\mathbb{1}(\cdot)$ denote an indicator function with

$$\mathbb{1}(x \in S) = \begin{cases} 1 & \text{if } x \in S \\ 0 & \text{otherwise.} \end{cases}$$

Let

$$\Pi_x(p) := F_x(p)(b - p).$$

Let

$$V_x(p) := \mathbb{E}[(p - c)\,\mathbb{1}(C < p) | X = x].$$

Some of our analysis will focus on a one-dimensional binary feature: $X = \{0, 1\}$, where $X = 1$ with probability $\theta$. In these cases, we simplify notation by parameterizing the principal's decision problem as that of choosing $\rho(0) = p_0$ and $\rho(1) = p_1$.

The principal's expected utility becomes

$$\Pi_{\text{rev}}(p_0, p_1) := (1 - \theta)\Pi_0(p_0) + \theta\Pi_1(p_1),$$

and the agent's utility becomes

$$V_{\text{rev}}(p_0, p_1) := (1 - \theta)V_0(p_0) + \theta V_1(p_1).$$

The principal moves first and chooses

$$p_0^*, p_1^* \in \arg\max_{p_0, p_1} \Pi_{\text{rev}}(p_0, p_1),$$

resulting in an equilibrium where the agent's utility is $V_{\text{rev}}(p_0^*, p_1^*)$.

Without loss of generality, we will refer to the situation when $X = 1$ as the "stronger" situation with generally lower cost for effort. That is, $p_0^* > p_1^*$.

Under garbling for binary $Y$, we simplify notation by parameterizing $\rho$ as $\rho(0) = p_0$ and $\rho(1) = p_1$. Let $p_0(\varepsilon), p_1(\varepsilon)$ denote the values of these parameters that maximize $\Pi_{\text{garb}}$ for a given $\varepsilon$.

## B  Proofs from Section 4

Here we give full proofs from Section 4.

### B.1  Proofs of Theorems 1 and 2

To prove Theorem 1, we first prove the following more detailed theorem.

THEOREM 3. *For any cost distribution that satisfies Assumptions 1, 2, if the cost is bounded above by $\bar{C}$ with $f(\bar{C}) > 0$, then there exists an instantiation of the principal's task value $b$ and a thresholding feature $X^t$ for which the agent prefers to reveal $X^t$. Specifically, the agent prefers to reveal $X^t$ if $b \in \left( \bar{C} + \frac{1 - F(t)}{f(\bar{C})}, t + \frac{F(t)}{f(t)} \right).$*

PROOF. For thresholding feature $X^t$, we have the scaled conditional distributions:

$$f_0(p) = \frac{f(p)}{1 - F(t)}\,\mathbb{1}(p > t); \quad f_1(p) = \frac{f(p)}{F(p)}\,\mathbb{1}(p \le t).$$

From these, we can compute that

$$F_0(p) = \int_0^p f_0(c)dc = \frac{F(p) - F(t)}{1 - F(t)}\,\mathbb{1}(p > t); \quad F_1(p) = \int_0^p f_1(c)dc = \frac{F(p)}{F(t)}\,\mathbb{1}(p \le t).$$

First, we show that for $b > \bar{C} + \frac{1 - F(t)}{f(\bar{C})}$, the optimal price in the high cost setting is $p_0^* = \bar{C}$. The principal's value is

$$\Pi_0(p) = F_0(p)(v - p) = \frac{F(p) - F(t)}{1 - F(t)}(v - p)\,\mathbb{1}(p > t).$$

Concavity of $F(p)$ in Assumption 1 for $p \ge p^*$ implies that $\Pi_0(p)$ is also concave for $p \ge p^*$. Since the function $p + \frac{F_0(p)}{f_0(p)}$ is monotone increasing (Assumption 2), $\Pi_0(p)$ is either maximized for $p_0^*$ satisfying first order condition $b = p_0^* + \frac{F_0(p_0^*)}{f_0(p_0^*)}$ if such a $p_0^*$ exists; or $p_0^* = \bar{C}$ if $b$ is sufficiently large. Specifically, $p_0^* = \bar{C}$ if $g_0^{-1}(b) > \bar{C}$, where $g_x(p) := p + \frac{F_x(p)}{f_x(p)}$. The lower bound on $b$ follows directly: $g_0^{-1}(b) > \bar{C} \iff b > \bar{C} + \frac{1 - F(t)}{f(\bar{C})}$.

Next, we show that for $b < t + \frac{F(t)}{f(t)}$, the price in the low cost setting is equal to price in the concealed setting: $p_1^* = p^*$. The principal's value is

$$\Pi_1(p) = F_1(p)(v - p) = \frac{F(p)}{F(t)}(v - p)\,\mathbb{1}(p \le t).$$

Concavity of $F(p)$ in Assumption 1 at $p^*$ implies that $\Pi_1(p)$ is concave near $p^*$ as long as $p^* < t$. This is sufficient for $\Pi_1(p)$ to be concave near its optimum $p_1^*$, since the first order condition for $\Pi_1(p)$ is identical to that of $\Pi(p)$. Again since $p + \frac{F_1(p)}{f_1(p)}$ is monotone increasing, $p_1^* = g_1^{-1}(b)$ if $g_1^{-1}(b) < t$; otherwise $p_1^* = t$. The upper bound on $b$ follows as $g_1^{-1}(b) < t \iff b < t + \frac{F(t)}{f(t)}$. Under this condition, the optimal price $p_1^* = g_1^{-1}(b) = g^{-1}(b) = p^*$.

Putting these together, when $b \in \left(\bar{C} + \frac{1-F(t)}{f(\bar{C})}, t + \frac{F(t)}{f(t)}\right)$, the price in the high cost setting is $p_0^* = \bar{C}$, and the price in the low cost setting is $p_1^* = p^*$. The total revealed agent value is $V_{\text{rev}}(p^*, \bar{C}) > V_{\text{rev}}(p^*, p^*) \implies V_{\text{rev}}(p^*, \bar{C}) > V_{\text{con}}(p^*)$.

To complete the proof, we show that there exists $t$ such that the set $\left(\bar{C} + \frac{1-F(t)}{f(\bar{C})}, t + \frac{F(t)}{f(t)}\right)$ is nonempty. The set is nonempty if

$$\bar{C} + \frac{1 - F(t)}{f(\bar{C})} < t + \frac{F(t)}{f(t)} \iff \bar{C} - t < -\frac{1}{f(\bar{C})} + \frac{F(t)}{f(\bar{C})} + \frac{F(t)}{f(t)} \iff \bar{C} - t < \frac{1}{f(\bar{C})}\left(\left(1 + \frac{f(\bar{C})}{f(t)}\right)F(t) - 1\right) \tag{1}$$

First, let $\delta_1 > 0$, and $\delta_1 f(\bar{C}) < 1$, and choose $t_1$ sufficiently large such that $\bar{C} - t_1 < \varepsilon$. Next, choose $\delta_2 > 0$ sufficiently small such that $\delta_1 < \frac{1}{f(\bar{C})}(1 - \delta_2)$, and choose $t_2$ sufficiently large such that $\left(\left(1 + \frac{f(\bar{C})}{f(t)}\right)F(t) - 1\right) > 1 - \delta_2$. Finally, choose $t_3 = \max(t_1, t_2)$. Then $t_3$ satisfies Equation (1). $\qed$

Intuitively, the specific bounds hold for $t$ sufficiently high relative to $b$: for a fixed $b$ not too much higher than $\bar{C}$, the bound $b < t + \frac{F(t)}{f(t)}$ holds for $t$ sufficiently high under Assumption 2. Similarly, the lower bound on $b$ is equivalent to $F(t) > 1 - f(\bar{C})(b - \bar{C})$, which also holds for $t$ sufficiently high. We summarize this more simply in Theorem 1 below.

**Theorem 1** (Agent prefers to reveal for high thresholds). *Under Assumptions 1 and 2, if the cost is bounded above by $\bar{C}$ with $f(\bar{C}) > 0$, and if $b \in \left(\bar{C}, \bar{C} + \frac{1}{f(\bar{C})}\right)$, then there exists a threshold $\underline{t}$ such that for all $t > \underline{t}$, the agent prefers to reveal $X^t$.*

PROOF. This proof follows directly from Theorem 3. Fix $b \in \left(\bar{C}, \bar{C} + \frac{1}{f(\bar{C})}\right)$. Choose $t_1$ sufficiently large that the lower bound holds: $b > \bar{C} + \frac{1-F(t_1)}{f(\bar{C})} \iff b - \bar{C} > \frac{1-F(t_1)}{f(\bar{C})}$. This is possible since $b > \bar{C}$ and $f(\bar{C}) > 0$. This lower bound will hold for any $t > t_1$.

Choose $t_2$ sufficiently large such that $b < t_2 + \frac{F(t_2)}{f(t_2)}$ the upper bound holds: $b < t_2 + \frac{F(t_2)}{f(t_2)}$. This is possible since $b < \max_{t \in [0,\bar{C}]} t + \frac{F(t)}{f(t)}$ (under Assumption 2). By Assumption 2, this upper bound will hold for any $t > t_2$. Choose $\underline{t} = \max(t_1, t_2)$. $\qed$

**Lemma 6.** *For $t \in \{0, \bar{C}\}$, $\Delta(t) = 0$.*

PROOF. When $t = 0$, $X^t = 0$ with probability 1. Therefore, $F_0(c) = F(c)$, $p_{0,t}^* = p^*$, and $\theta = 1$, so $V_{\text{rev}}^t(p_{1,t}^*, p_{0,t}^*) = V_{\text{con}}(p^*)$ (for any $p_{1,t}^*$). A similar argument holds for $t = \bar{C}$. $\qed$

**Theorem 2** (Agent prefers to conceal for low thresholds). *Under Assumptions 1, 2, and 3, if $f(0) > 0$, then $\Delta'(0) < 0$.*

PROOF. For notational convenience, let the superscript $t$ represent all principal and agent values defined in Section 3.1, but with $X = X^t$:

$$F_0^t(c) = P(C \le c | X^t = 0), \quad F_1^t(c) = P(C \le c | X^t = 1), \text{etc.}$$

Let $p_0(t), p_1(t)$ be defined as the principal's optimal prices under revealed $X^t$,

$$p_0(t), p_1(t) \in \underset{p_0, p_1}{\arg\max} \Pi_{\text{rev}}^t(p_0, p_1).$$

With a bit of abuse of notation, let $V_{\text{rev}}(t) := V_{\text{rev}}^t(p_0(t), p_1(t))$. Then $\Delta(t) = V_{\text{rev}}(t) - V_{\text{con}}(p^*)$. Since $V_{\text{con}}(p^*)$ does not depend on $t$, we need only show that $V_{\text{rev}}'(0) < 0$. Differentiating this, we have

$$V_{\text{rev}}'(0) = \frac{\partial}{\partial t}\left(V_1^t(p_1(t))F(t)\right)\Big|_{t=0} + \frac{\partial}{\partial t}\left(V_0^t(p_0(t))(1 - F(t))\right)\Big|_{t=0}$$

The first term is 0:

$$\frac{\partial}{\partial t}\left(V_1^t(p_1(t))F(t)\right)\Big|_{t=0} = F(0)\frac{\partial}{\partial t}V_1^t(p_1(t))\Big|_{t=0} + f(0)V_1^t(p_1(t))\Big|_{t=0} = 0$$

Note that $\lim_{t \to 0} p_1(t) = 0$, which implies that $V_1^t(p_1(t))\big|_{t=0} = 0$. Thus,

$$V_{\text{rev}}'(0) = \frac{\partial}{\partial t}\left(V_0^t(p_0(t))(1 - F(t))\right)\Big|_{t=0} = (1 - F(0))\frac{\partial}{\partial t}V_0^t(p_0(t))\Big|_{t=0} - f(0)V_0^t(p_0(t))\Big|_{t=0} = \frac{\partial}{\partial t}V_0^t(p_0(t))\Big|_{t=0} - f(0)V_{\text{con}}(p^*) \tag{2}$$

We solve for the above using the chain rule, substituting $p = p_0(t)$:

$$\frac{\partial}{\partial t}V_0^t(p_0(t))\Big|_{t=0} = \frac{\partial}{\partial p}V_0^t(p)\Big|_{t=0} p_0'(0) + \frac{\partial}{\partial t}V_0^t(p)\Big|_{t=0} \tag{3}$$

Solving for each of the terms in this expression:

$$\frac{\partial}{\partial p} V_0^t(p)\Big|_{t=0} = F_0^t(p)\Big|_{t=0} = F(p^*) \tag{4}$$

$$\frac{\partial}{\partial t} V_0^t(p)\Big|_{t=0} = p^* \frac{\partial}{\partial t} F_0^t(p)\Big|_{t=0} - \frac{\partial}{\partial t} \int_0^p c f_0^t(c) dc \Big|_{t=0}$$

$$\frac{\partial}{\partial t} F_0^t(p)\Big|_{t=0} = \frac{\partial}{\partial t} \frac{F(p) - F(t)}{1 - F(t)}\Big|_{t=0} = \frac{f(t)(F(p) - 1)}{(1 - F(t))^2}\Big|_{t=0} = f(0)(F(p^*) - 1)$$

$$\frac{\partial}{\partial t} \int_0^p c f_0^t(c) dc \Big|_{t=0} = E[\mathbb{1}(C < p^*)C] f(0)$$

$$\implies \frac{\partial}{\partial t} V_0^t(p)\Big|_{t=0} = f(0) \left( p^*(F(p^*) - 1) - E[\mathbb{1}(C < p^*)C] \right) = f(0) \left( V(p^*) - p^* \right) \tag{5}$$

Finally, to solve for $p_0'(t)$, we either have $p_0'(t) = 0$ for $b >> C$, in which case the proof is complete and it immediately follows that $\Delta'(0) < 0$; or, we suppose $p_0(t)$ satisfies the first order condition: $p_0(t) + \frac{F(p_0(t)) - F(t)}{f(p_0(t))} = v$. Differentiating on both sides of this first order condition, we have

$$p_0'(t) = \frac{f(t)f(p_0(t))}{2f(p_0(t))^2 - F(p_0(t))f'(p_0(t)) + F(t)f'(p_0(t))} \implies p_0'(0) = \frac{f(0)}{2f(p^*) - \frac{F(p^*)}{f(p^*)}f'(p^*)} \tag{6}$$

We now combine Equations (4), (5), and (6) with Equation (3) to get that

$$\frac{\partial}{\partial t} V_0^t(p_0(t))\Big|_{t=0} = f(0) \left( \frac{F(p^*)}{2f(p^*) - \frac{F(p^*)f'(p^*)}{f(p^*)}} - \left( V_{\text{con}}(p^*) - p^* \right) \right).$$

Putting this all together with Equation (2), $V_{\text{rev}}'(0) < 0$ if and only if

$$f(0) \left( \frac{F(p^*)}{2f(p^*) - \frac{F(p^*)f'(p^*)}{f(p^*)}} - \left( V_{\text{con}}(p^*) - p^* \right) - V_{\text{con}}(p^*) \right) < 0$$

$$\iff 1 - \frac{F(p^*)}{p^* f(p^*)} + \frac{\partial}{\partial p} \frac{F(p)}{f(p)}\Big|_{p=p^*} > 0$$

This holds by Assumptions 2 and 3. Therefore, $\Delta'(0) < 0$. □

## B.2  Proofs from Section 4.2

**Lemma 1.** Total welfare increases under revelation $(W_{\text{rev}}(\rho^*) - W_{\text{con}}(p^*))$ only if task completion quantity increases under revelation,

$$F(p^*) < \mathbb{E}[\mathbb{1}(C < \rho^*(X)].$$

PROOF. Let $\underline{X} \subseteq X$ be the set of $x$ values for which $\rho(x) < p^*$, and let $\overline{X} = X \setminus \underline{X}$.

$$W_{\text{rev}}(\rho^*) - W_{\text{con}}(p^*) = \mathbb{E}[\mathbb{1}(C < \rho^*(X))(b - C)] - \mathbb{E}[\mathbb{1}(C < p^*)(b - C)]$$

$$= \mathbb{E}[\mathbb{1}(C < \rho^*(X))(b - C)|X \in \underline{X}]\mathbb{P}(X \in \underline{X}) + \mathbb{E}[\mathbb{1}(C < \rho^*(X))(b - C)|X \in \overline{X}]\mathbb{P}(X \in \overline{X}) - \mathbb{E}[\mathbb{1}(C < p^*)(b - C)]$$

$$= \mathbb{E}[(\mathbb{1}(C < \rho^*(X)) - \mathbb{1}(C < p^*))(b - C)|X \in \overline{X}]\mathbb{P}(X \in \overline{X})$$

$$- \mathbb{E}[(\mathbb{1}(C < p^*) - \mathbb{1}(C < \rho^*(X)))(b - C)|X \in \underline{X}]\mathbb{P}(X \in \underline{X})$$

$$\leq \mathbb{E}[(\mathbb{1}(C < \rho^*(X)) - \mathbb{1}(C < p^*))(b - p^*)|X \in \overline{X}]\mathbb{P}(X \in \overline{X})$$

$$- \mathbb{E}[(\mathbb{1}(C < p^*) - \mathbb{1}(C < \rho^*(X)))(b - p^*)|X \in \underline{X}]\mathbb{P}(X \in \underline{X})$$

$$\implies W_{\text{rev}}(\rho^*) - W_{\text{con}}(p^*) \leq (b - p^*)(\mathbb{E}[\mathbb{1}(C < \rho^*(X)] - \mathbb{E}[\mathbb{1}(C < p^*)])$$

Since $(b - p^*) > 0$, if $(\mathbb{E}[\mathbb{1}(C < \rho^*(X)] - \mathbb{E}[\mathbb{1}(C < p^*)]) \leq 0$, then $W_{\text{rev}}(\rho^*) - W_{\text{con}}(p^*) \leq 0$. □

## C  Proofs from Section 5

Here we give proofs for results for the garbling model presented in Section 5.

## C.1 Proofs from Section 5.4

**Lemma 5** (Privacy of garbling mechanism). *For $\theta \in (0,1)$, for all $\varepsilon \leq O(\theta)$, the garbling mechanism is an $O(\varepsilon)$-local randomizer with respect to $X$.*

PROOF. Suppose without loss of generality that $\theta < \frac{1}{2}$ (i.e., $X = 0$ with higher probability).

$$\max_{y,x \in \{0,1\}} \frac{\mathbb{P}(Y = y | X = x)}{\mathbb{P}(Y = y | X = 1 - x)} \leq \frac{\mathbb{P}(Y = 1 | X = 1)}{\mathbb{P}(Y = 1 | X = 0)} = \frac{\theta(1 - \varepsilon) + \varepsilon}{\theta(1 - \varepsilon)} \leq e^{O(\varepsilon)}$$

The last inequality holds for $\varepsilon$ sufficiently small: $\varepsilon \leq \frac{\theta}{\theta+1} \implies \frac{\varepsilon}{\theta(1-\varepsilon)} \leq 1$.

A pricing mechanism that accesses $X$ only through this garbling mechanism is therefore locally differentially private with a corresponding privacy parameter. □

## C.2 Proofs from Section 5.3

**Lemma 3** (Reducing garbling initially increases total welfare). *Relative to full concealment with $\varepsilon = 0$, revealing $Y$ with some garbled noise initially does not decrease total welfare: $W'_{\text{garb}}(0) \geq 0$. The inequality is strict if $\Pi'_{\text{garb}}(0) > 0$, which is true under strict concavity of $\Pi_0(p), \Pi_1(p)$ (Assumption 4) and the MLRP (Assumption 6).*

PROOF.

$$W'_{\text{garb}}(\varepsilon) = V'_{\text{garb}}(\varepsilon) + \Pi'_{\text{garb}}(\varepsilon)$$

$V'_{\text{garb}}(0) = 0$, so $W'_{\text{garb}}(0) \geq 0$. The strict inequality comes from applying Lemma 13 that $\Pi'_{\text{garb}}(0) > 0$. □

**Lemma 2** (Optimal garbling increases welfare over concealment). *Let $\varepsilon^* \in \arg\max_{\varepsilon \in [0,1]} V_{\text{garb}}(\varepsilon)$. Then $W_{\text{garb}}(\varepsilon^*) \geq W_{\text{garb}}(0)$.*

PROOF.

$$W_{\text{garb}}(\varepsilon^*) = V_{\text{garb}}(\varepsilon^*) + \Pi_{\text{garb}}(\varepsilon^*)$$

Lemma 12 implies $\Pi_{\text{garb}}(\varepsilon^*) \geq \Pi_{\text{garb}}(0)$. By optimality of $\varepsilon^*$, $V_{\text{garb}}(\varepsilon^*) \geq V_{\text{garb}}(0)$. □

**Lemma 4** (More information increases total welfare relative to agent optimal garbling). *$W'_{\text{garb}}(\varepsilon) \geq V'_{\text{garb}}(\varepsilon)$ for all $\varepsilon \in [0, 1]$. The inequality is strict if $\Pi'_{\text{garb}}(0) > 0$, which is true under strict concavity of $\Pi_0(p), \Pi_1(p)$ (Assumption 4) and the MLRP (Assumption 6).*

PROOF. Lemma 12 implies that $\Pi'_{\text{garb}}(\varepsilon) > 0$, which implies that $W'_{\text{garb}}(\varepsilon) \geq V'_{\text{garb}}(\varepsilon)$. The inequality is strict under the conditions of Lemma 13. □

# D Additional Results on Welfare Effects of Information Revelation

## D.1 Uniform Simulation for Thresholding Features

To illustrate Theorems 1 and 2 on thresholding features, let $C \sim \text{Unif}(0, 1)$. Let $X^t$ be a thresholding feature for $C$, where $X^t = 1$ if $C \leq t$, and 0 otherwise. Figure 5 shows the agent's utility difference between revealing and concealing $X^t$ for all pairs of $b \in (0, 2), t \in (0, 1)$.

## D.2 Agent's Revelation Incentives: General Conditions

We give more general conditions the agent to prefer to reveal or conceal for $X$ beyond thresholding features.

We first show that the agent prefers the hidden information setting if the cost conditional on $X$ is close to zero for some values of $X$, and still not very high for other values of $X$. Theorem 1 shows a general hazard rate condition that leads to this conclusion.

Next, we give sufficient conditions for the agent to conceal and reveal under general $X$, applying identities related to the analysis done by Aguirre et al. [4].

*D.2.1 Concealment condition with one zero-cost type.* We present a sufficient condition for the agent to prefer to conceal $X$ when one of the agent types is anchored at zero. That is, $X = 1$ implies that the agent incurs zero cost. Proposition 1 gives a sufficient condition on $F_0$ for the agent to prefer for the environmental variable $X$ to remain concealed.

**Proposition 1** (Sufficient concealment condition with zero-cost type). *Suppose $F_0$ is a concave and continuously differentiable CDF. Suppose $C | X = 1$ takes value 0 with probability 1. Suppose the ratio $\frac{F_0(p)}{f_0(p)}$ is strictly monotone increasing for $p > 0$. Then $V_{\text{con}}(p^*) > V_{\text{rev}}(p_0^*, p_1^*)$ if*

$$\theta > (1 - \theta)\frac{1}{\eta((1 - \theta)p_0^*)} - \frac{1}{\eta_0\left(p_0^*\right)}, \tag{7}$$

*where $\eta(p) = \frac{p(1-\theta)f_0(p)}{(1-\theta)F_0(p)+\theta}$ and $\eta_0(p) = \frac{pf_0(p)}{F_0(p)}$ are the respective price elasticities for task completion quantity for the mixture distribution $C$ and the conditional distribution $F_0$.*

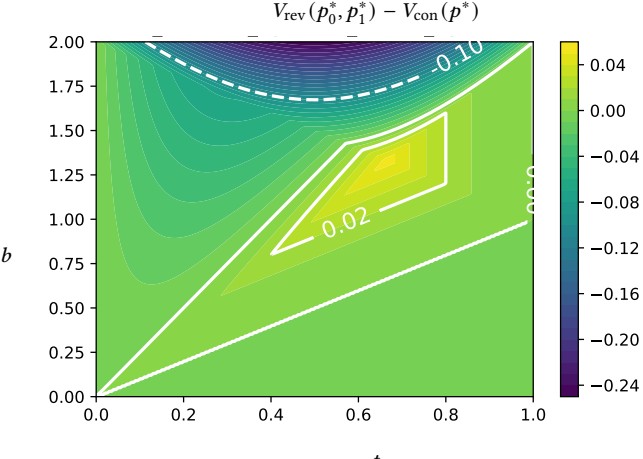

**Figure 5: Difference between agent's utility in the revealed setting and concealed settings when $C \sim \text{Unif}(0, 1)$, and $X = X^t$ is a thresholding feature. For a given pair $(b, t)$, a positive utility difference means the agent prefers to reveal $X^t$, and a negative utility difference means the agent prefers to conceal $X^t$. The agent always prefers to conceal for $t$ sufficiently close to 0, which matches Theorem 2. For $b \in (1, 2)$, the agent prefers to reveal for $t > \frac{b}{2}$, which matches Theorem 1. At $b = 2$, the agent always prefers to conceal. For $b < 1$, the agent actually prefers to conceal for sufficiently high $t$.**

Intuitively, the elasticity $\eta$ captures the sensitivity of task completion to the offered price. Thus, the inequality in equation (7) corresponds to a scenario when the sensitivity of the task completion to price when $X = 0$ does not differ too strongly from that of the concealed setting. For example, this arises when $F_0$ is close to the constant function 1. We give another example using the exponential distribution in Section D.3 below, where equation (7) holds if the mean of $F_0$ is low enough. In summary, the agent prefers concealment if the higher cost type still has relatively low cost.

*D.2.2 Concealment and revelation conditions under a decreasing ratio assumption.* To give additional sufficient conditions for concealment and revelation for variables more general than thresholding features, we apply an analysis technique similar to that of Aguirre et al. [4], who analyzed the effects of third degree monopoly price discrimination on total welfare.

Suppose the principal, on knowing $X$, is constrained to choose transfers $p_0, p_1$ subject to the constraint that $p_0 - p_1 < r$ for some $r \geq 0$. Let $p_0(r), p_1(r)$ denote the principal's optimal transfers under this constraint:

$$p_0(r), p_1(r) \in \arg\max_{p_0, p_1} \ \Pi_{\text{rev}}(p_0, p_1)$$
$$\text{s.t.} \quad p_0 - p_1 \geq r. \tag{8}$$

For notational convenience, let $V_{\text{const}}(r) := V_{\text{rev}}(p_0(r), p_1(r))$. In a similar structure to Aguirre et al. [4], the results in this section come from considering the "marginal effect of relaxing the constraint" on the agent's value.

Under the following closely analogous assumptions to those invoked by Aguirre et al. [4], we derive properties of $V_{\text{const}}(r)$.

**Assumption 4** (Concave principal utility). *The principal's utility in each realized environment is strictly concave: $\Pi_0''(p) < 0$, $\Pi_1''(p) < 0$.*

**Assumption 5** (Decreasing ratio condition (DRC)). *The ratios $\frac{V_0'(p)}{\Pi_0''(p)}$ and $\frac{V_1'(p)}{\Pi_1''(p)}$ are both decreasing in $p$.*

Assumption 5 is analogous to the "increasing ratio condition" assumption from Aguirre et al. [4], which instead has the derivative of total welfare in the numerator. Our analysis naturally extends this to focus on agent utility. Assumption 5 holds in almost the same set of conditions as the assumption on total welfare from Aguirre et al. [4], and we discuss the subtleties of the differences between these assumptions in Appendix F.3.

**Lemma 7.** *Under Assumptions 4 and 5, $V_{\text{const}}(r)$ is strictly quasi-convex for $r \in [0, p_0^* - p_1^*]$. That is, if there exists $\hat{r} \in [0, p_0^* - p_1^*]$ such that $V_{\text{const}}'(\hat{r}) = 0$, then $V_{\text{const}}''(\hat{r}) > 0$.*

The strict quasi-convexity of the agent's utility in $r$ makes it possible to derive sufficient conditions for revelation and concealment by differentiating $V_{\text{const}}$ and evaluating the sign of the derivative at extreme values of $r$. Adapting this machinery from Aguirre et al. [4], but focusing on the agent's value instead of total welfare, we give such sufficient conditions for the agent to prefer concealing or revealing $X$ below.

**Proposition 2** (Sufficient concealment condition under DRC). *Under Assumptions 4 and 5, $V_{con}(p^*) > V_{rev}(p_0^*, p_1^*)$ if*

$$\frac{(1-\theta)(b-p_0^*)}{2-\sigma_0(p_0^*)} < \frac{\theta(b-p_1^*)}{2-\sigma_1(p_1^*)}, \tag{9}$$

*where $\sigma_x(p) = \frac{F_x(p)f_x'(p)}{f_x^2(p)}$ is the curvature of the inverse of the task completion quantity function $F_x(p)$.*

Proposition 2 implies that a high enough difference in curvature between $\sigma_0(p_0^*)$ and $\sigma_1(p_1^*)$ implies that the agent will prefer the concealed contract over the revealed contract. That is, inverse task completion quantity when $X = 0$ is more convex than the inverse task completion quantity when $X = 1$ at the revealed transfers $p_0^*, p_1^*$. This is exactly the flipped version of the condition in Aguirre et al. [4]'s Proposition 2, which implied that total welfare is higher under price discrimination. We next give a sufficient condition for the agent to prefer revelation.

**Proposition 3** (Sufficient revelation condition under DRC). *Under Assumptions 4 and 5, $V_{con}(p^*) < V_{rev}(p_0^*, p_1^*)$ if*

$$\frac{2 + L(p^*)\alpha_1(p^*)}{2 + L(p^*)\alpha_0(p^*)} > \frac{\theta F_1(p^*)/f_1(p^*)}{(1-\theta)F_0(p^*)/f_0(p^*)}, \tag{10}$$

*where $L(p) = \frac{b-p}{p}$ is the Lerner index [35], and $\alpha_x(p) = \frac{-pf_x'(p)}{f_x(p)}$ is the curvature of the task completion quantity function $F_x(p)$,*

Intuitively, Proposition 3 says that if the curvatures of $F_0$ and $F_1$ are different enough (relative to the ratio of the CDFs themselves), then the agent will prefer to reveal the environmental variable $X$.

*Remark.* An important property analyzed by Milgrom [39] is the monotone likelihood ratio property (MLRP), which here would say that $\frac{f_0(c)}{f_1(c)}$ is increasing for all $c$. The MLRP would imply that both sides of the inequality in equation (10) are greater than 1. However, this does not necessarily imply an order between these ratios, and there exist distributions that satisfy the MLRP that yield either of the inequality directions above. We give a specific example of this using the Weibull distribution in Section D.3 below.

## D.3 Example: Exponential and Weibull Distributions

To concretely illustrate the conditions in Propositions 1, 2, and 3, we parameterize the conditional cost distributions using the exponential distribution and the more general Weibull distribution.

First, to illustrate the condition in Proposition 1, let $C|X = 0 \sim \text{Exp}(\frac{1}{\lambda_0})$, where $\lambda_0$ represents the scale parameter and is also the mean of the distribution. Specifically,

$$F_0(c) = \begin{cases} 1 - e^{-\frac{1}{\lambda_0}c} & c \geq 0 \\ 0 & c < 0. \end{cases} \tag{11}$$

Then the condition in Equation (7) is equivalent to $\lambda_0 < b\psi(\theta)$, where $\psi(\theta) = \frac{\theta}{\left(\left(\frac{1}{1-\theta}\right)^{\frac{1}{\theta}} - \left(\frac{1}{1-\theta}\right)^{\frac{1-\theta}{\theta}} - \theta\right)}$ is monotone decreasing function bounded between 0 and 1 for $\theta \in [0, 1]$. Thus, as long as the average cost $\lambda_0$ is less than a $\theta$-dependent scaling of the task completion value $b$, the condition in Proposition 1 holds. In other words, if the average cost when $X = 0$ is not too high, then the agent will prefer concealment.

Beyond fixing $F_1$ at zero cost, let $C$ be a mixture of exponential distributions with $C|X = 0 \sim \text{Exp}(\frac{1}{\lambda_0})$ and $C|X = 1 \sim \text{Exp}(\frac{1}{\lambda_1})$, where

$$F_x(c) = \begin{cases} 1 - e^{-\frac{1}{\lambda_x}c} & c \geq 0 \\ 0 & c < 0. \end{cases} \tag{12}$$

Figure 6 plots the difference $V_{rev}(p_0^*, p_1^*) - V_{con}(p^*)$ for all $\lambda_0, \lambda_1 \in [0, b]$. As seen in Proposition 1, if $\lambda_1 = 0$, then the agent prefers to hide if $\lambda_0$ is sufficiently low. This continues to hold for $\lambda_1$ sufficiently close to 0. More generally, Figure 6 shows that the agent prefers to reveal if the means $\lambda_0, \lambda_1$ are sufficiently far apart.

For the exponential mixture, the inequalities in Propositions 2 and 3 do not hold for any combinations of $\lambda_0, \lambda_1$. Thus, the condition in Proposition 1 covers cases not covered by Proposition 2. However, we see Propositions 2 and 3 take effect for the more general Weibull distribution, with

$$F_x(c) = \begin{cases} 1 - e^{\left(-\frac{1}{\lambda_x}c\right)^{k_x}} & c \geq 0 \\ 0 & c < 0. \end{cases} \tag{13}$$

For example, for a fixed $\lambda_0, \lambda_1$, increasing $k = k_0 = k_1$ increases the difference in curvature between $F_0$ and $F_1$ at $p^*$, yielding a set of values of $k > 1$ in which the condition in Proposition 3 holds. For Proposition 2, the condition holds if $k_1$ is sufficiently small, and $k_0$ is sufficiently large for $\lambda_0 > \lambda_1$.

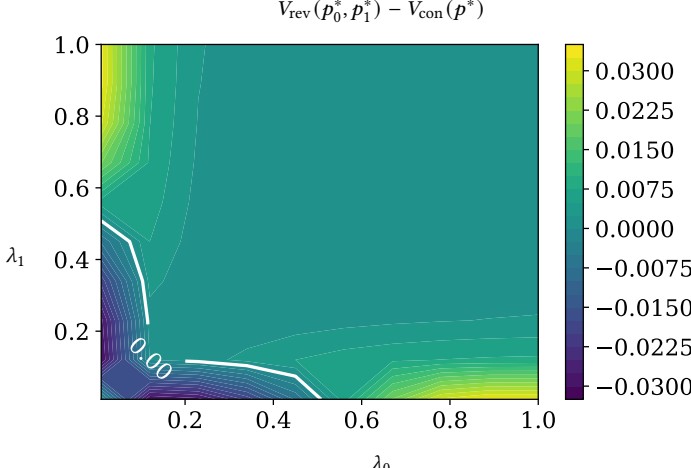

**Figure 6: Difference between agent's utility in the revealed setting and concealed settings when $C$ is distributed as a mixture of exponentials. For each pair $(\lambda_0, \lambda_1)$, a positive value indicates that the agent prefers revelation, and a negative value indicates that the agent prefers concealment. The contour line shows all $(\lambda_0, \lambda_1)$ for which $V_{rev}(p_0^*, p_1^*) - V_{con}(p^*) = 0$. The parameters $b = 1$ and $\theta = \frac{1}{2}$ are fixed, and $\lambda_0$ and $\lambda_1$ are varied up to $b$.**

### D.4 Principal's Revelation Preferences

While the agent might sometimes prefer the hidden setting over the revealed setting, we next show that the principal always prefers revelation. First, Lemma 8 shows that the principal is never worse off under revelation.

**Lemma 8** (Principal prefers revelation). *Revealing $X$ never decreases the value of the principal: $\Pi_{rev}(\rho^*) \geq \Pi_{con}(p^*)$, where $\rho^* \in \arg\max_\rho \Pi_{rev}(\rho)$. Revealing $X$ strictly increases the value of the principal only if $X$ and $C$ are not independent.*

The principal strictly benefits from information revelation if the monotone likelihood ratio property (MLRP) is satisfied between the revealed distributions.

**Assumption 6** (Monotone likelihood ratio property (MLRP) [39]). *The ratio $\frac{f_0(c)}{f_1(c)}$ is strictly increasing in $c$.*

**Lemma 9** (Principal strictly benefits from revelation). *Let $F_0$ and $F_1$ be continuously differentiable CDFs. If the MLRP holds (Assumption 6), then the principal strictly benefits when $X$ is revealed: $\Pi_{rev}(p_0^*, p_1^*) > \Pi_{con}(p^*)$.*

Intuitively, as the first mover, the principal will never be hurt by having additional freedom to condition on $X$ when selecting prices that maximize their utility. Lemma 9 gives the MLRP as a sufficient condition for revelation of $X$ to yield a strict benefit for the principal.

## E  Example Where Total Welfare Decreases under Revelation

It is still possible for total welfare to decrease under information revelation. Mirroring an example from Varian [48], we provide an illustrative example here where total welfare decreases when task completion quantity does not increase.

Suppose $C|X = 1 \sim \text{Unif}(0, 1)$, and $C|X = 0 \sim \text{Unif}(\frac{1}{2}, \frac{3}{2})$. Suppose $\theta = \frac{1}{2}$. Suppose $b = 1$. Then the optimal payments for each of $F, F_0, F_1$ all fall in the "interior" of $F(x)$:

$$\frac{1}{2} \leq p_1^* < p^* < p_0^* \leq 1.$$

When the solutions all fall in the interior, we have $p^* = \frac{p_1^* + p_0^*}{2}$, and $F(p^*) = \frac{F_1(p_1^*) + F_0(p_0^*)}{2}$. This now violates the necessary condition in Lemma 1, since the output does not increase under revelation, but the payments change. Total welfare decreases as long as $p_0^* \neq p_1^*$.

## F  Proofs from Section D

Here we give full proofs from Section D.

### F.1  Proof of Proposition 1

To prove Proposition 1, we first reorganize the difference between the agent's concealed and revealed utilities:

$$V_{con}(p^*) - V_{rev}(p_1^*, p_1^*) = \theta \Delta V_1 - (1 - \theta) \Delta V_0$$

where
$$\Delta V_0 := V_0(p_0^*) - V_0(p^*); \quad \Delta V_1 := V_1(p^*) - V_1(p_1^*).$$

We begin with Lemma 10 below which upper bounds $\Delta V_0$.

**Lemma 10.** For any concave and continuously differentiable CDF $F_0$, $\Delta V_0 \leq p_0^* - \bar{p}$ for any $\bar{p} < p_0^*$.

PROOF. $\Delta V_0 \leq p_0^* - \bar{p}$ if
$$\frac{V_0(p_0^*) - V_0(\bar{p})}{p_0^* - \bar{p}} \leq 1.$$

We upper bound this difference by differentiating $V_0$:
$$V_0'(p) = \frac{d}{dp} \int_0^p (p - c) f_0(c) dc = F_0(p)$$

Since $F_0$ is a concave and continuously differentiable CDF, by the mean value theorem,
$$\frac{V_0(p_0^*) - V_0(\bar{p})}{p_0^* - \bar{p}} \leq \sup_p V_0'(p) = \sup_p F_0(p) \leq 1.$$

□

We now leverage Lemma 10 to prove the full proposition.

**Proposition 1** (Sufficient concealment condition with zero-cost type). Suppose $F_0$ is a concave and continuously differentiable CDF. Suppose $C|X = 1$ takes value 0 with probability 1. Suppose the ratio $\frac{F_0(p)}{f_0(p)}$ is strictly monotone increasing for $p > 0$. Then $V_{\text{con}}(p^*) > V_{\text{rev}}(p_0^*, p_1^*)$ if
$$\theta > (1 - \theta) \frac{1}{\eta((1 - \theta) p_0^*)} - \frac{1}{\eta_0 \left( p_0^* \right)}$$

where $\eta(p) = \frac{p(1-\theta)f_0(p)}{(1-\theta)F_0(p)+\theta}$ and $\eta_0(p) = \frac{pf_0(p)}{F_0(p)}$ are the respective price elasticities for task completion quantity for the mixture distribution $C$ and the conditional distribution $C|X = 0$.

PROOF. We consider the extreme case where $C|X = 1$ has value 0 with probability 1. For this distribution of $C|X = 1$, we show that Equation (7) implies that
$$\theta \Delta V_1 > (1 - \theta) \Delta V_0. \tag{14}$$

First, for any nonzero $C|X = 0$, we have that $p^* < p_0^*$. Thus, Lemma 10 gives that
$$(1 - \theta) \Delta V_0 \leq (1 - \theta)(p_0^* - p^*).$$

Next, we further upper bound this by showing that for any $F_0$ that satisfies Equation (7),
$$(1 - \theta)(p_0^* - p^*) < \theta p^*. \tag{15}$$

Equation (14) then follows from the fact that $\Delta V_1 = p^*$ when $C|X = 1$ is always zero.

We now prove that equation (15) holds under equation (7). First, note that
$$(1 - \theta)(p_0^* - p^*) < \theta p^* \iff (1 - \theta) p_0^* < p^*.$$

Since $F_0$ is concave and continuously differentiable, $p^*$ satisfies the following first-order condition:
$$p^* + \frac{(1 - \theta) F_0(p^*) + \theta}{(1 - \theta) f_0(p^*)} = b. \tag{16}$$

Since $\frac{F_0(p)}{f_0(p)}$ is strictly monotone increasing for $p > 0$ and $F_0(p)$ is concave, $p + \frac{(1-\theta)F_0(p)+\theta}{(1-\theta)f_0(p)}$ is also strictly monotone increasing for $p > 0$. Therefore, $(1 - \theta) p_0^* < p^*$ if and only if
$$(1 - \theta) p_0^* + \frac{(1 - \theta) F_0((1 - \theta) p_0^*) + \theta}{(1 - \theta) f_0((1 - \theta) p_0^*)} < b.$$

We also have that $p_0^*$ satisfies the first-order condition
$$p_0^* + \frac{F_0(p_0^*)}{f_0(p_0^*)} = b.$$

Therefore, $(1 - \theta) p_0^* < p^*$ if and only if
$$(1 - \theta) p_0^* + \frac{(1 - \theta) F_0((1 - \theta) p_0^*) + \theta}{(1 - \theta) f_0((1 - \theta) p_0^*)} < p_0^* + \frac{F_0(p_0^*)}{f_0(p_0^*)},$$

which is equivalent to the condition in equation (7).

$\square$

## F.2 Proofs for Propositions 2 and 3

Here we give proofs for Propositions 2 and 3. These analyses parallel those of Aguirre et al. [4] for the effects of price discrimination on total welfare.

We first prove Lemma 7, which follows from Assumption 4 and 5.

**Lemma 7.** Under Assumptions 4 and 5, $V_{\text{const}}(r)$ is strictly quasi-convex for $r \in [0, p_0^* - p_1^*]$. That is, if there exists $\hat{r} \in [0, p_0^* - p_1^*]$ such that $V_{\text{const}}(\hat{r}) = 0$, then $V_{\text{const}}''(\hat{r}) > 0$.

PROOF. The constraint in equation (8) is binding when $r \in [0, p_0^* - p_1^*]$. Therefore, the optimization problem in equation (8) can be rewritten as

$$\max_{p_1} \ \Pi_1(p_1) + \Pi_0(p_1 + r),$$

yielding a first-order condition that $\Pi_1'(p_1) + \Pi_0'(p_1 + r) = 0$. Further differentiating this first-order condition, as done by Aguirre et al. [4], yields that

$$p_1'(r) = \frac{-\Pi_0''(p_0(r))}{\Pi_0''(p_0(r)) + \Pi_1''(p_1(r))}.$$

A similar method shows that

$$p_0'(r) = \frac{\Pi_1''(p_1(r))}{\Pi_0''(p_0(r)) + \Pi_1''(p_1(r))}.$$

Thus, we have that

$$
\begin{aligned}
V_{\text{const}}'(r) &= (1-\theta)V_0(p_0(r))p_0'(r) + \theta V_1(p_1(r))p_1'(r) \\
&= \left( \frac{-\Pi_1''(p_1(r)\Pi_0''(p_0(r)))}{\Pi_0''(p_0(r)) + \Pi_1''(p_1(r))} \right) \left( \frac{\theta V_1'(p_1(r))}{\Pi_1''(p_1(r))} - \frac{(1-\theta)V_0'(p_0(r))}{\Pi_0''(p_0(r))} \right) \\
&= \left( \frac{-\Pi_1''(p_1(r)\Pi_0''(p_0(r)))}{\Pi_0''(p_0(r)) + \Pi_1''(p_1(r))} \right) (\theta w_1(p_1(r)) - (1-\theta)w_0(p_0(r))),
\end{aligned}
\tag{17}
$$

where $w_x(p) := \frac{V_x'(p)}{\Pi_x''(p)}$.

Taking the second derivative, we have that

$$
\begin{aligned}
V_{\text{const}}''(r) &= \left( \frac{-\Pi_1''(p_1(r)\Pi_0''(p_0(r)))}{\Pi_0''(p_0(r)) + \Pi_1''(p_1(r))} \right) (\theta w_1'(p_1(r))p_1'(r) - (1-\theta)w_0'(p_0(r))p_0'(r)) \\
&\quad + (\theta w_1(p_1(r)) - (1-\theta)w_0(p_0(r)))\frac{\partial}{\partial r} \left( \frac{-\Pi_1''(p_1(r)\Pi_0''(p_0(r)))}{\Pi_0''(p_0(r)) + \Pi_1''(p_1(r))} \right).
\end{aligned}
$$

The first term $\left( \frac{-\Pi_1''(p_1(r)\Pi_0''(p_0(r)))}{\Pi_0''(p_0(r)) + \Pi_1''(p_1(r))} \right)$ is positive by strict concavity given by Assumption 4.

If $V_{\text{const}}'(\hat{r}) = 0$, then $\theta w_1(p_1(\hat{r})) - (1-\theta)w_0(p_0(\hat{r})) = 0$. By the DRC, $w_1'(p_1(\hat{r}))p_1'(\hat{r}) > 0$ since $w_1'(p_1(\hat{r})) < 0$ and $p_1'(\hat{r}) < 0$. Similarly, $w_0'(p_0(\hat{r}))p_0'(\hat{r}) < 0$. Therefore, $V_{\text{const}}''(\hat{r}) > 0$.

$\square$

Given Lemma 7, we now prove Propositions 2 and 3 by signing the derivative $V_{\text{const}}'(r)$ for extreme values of $r$.

**Proposition 2** (Sufficient concealment condition under DRC). Under Assumptions 4 and 5, $V_{\text{con}}(p^*) > V_{\text{rev}}(p_0^*, p_1^*)$ if

$$\frac{(1-\theta)(b - p_0^*)}{2 - \sigma_0(p_0^*)} < \frac{\theta(b - p_1^*)}{2 - \sigma_1(p_1^*)},$$

where $\sigma_x(p) = \frac{F_x(p)f_x'(p)}{f_x^2(p)}$ is the curvature of the inverse of the task completion quantity function $F_x(p)$.

PROOF. If $V_{\text{const}}(r)$ is strictly monotone decreasing in $r$, then $V_{\text{con}}(p^*) > V_{\text{rev}}(p_0^*, p_1^*)$. Since $V_{\text{const}}(r)$ is strictly quasi-convex, a sufficient condition for $V_{\text{const}}(r)$ to be strictly monotone decreasing is $V_{\text{const}}'(p_0^* - p_1^*) < 0$.

By equation (17), we have that $V_{\text{const}}'(p_0^* - p_1^*) < 0$ if $\theta w_1(p_1^*) - (1-\theta)w_0(p_0^*) < 0$. By the first-order condition that

$$b - p_x^* = \frac{F_x(p_x^*)}{f_x(p_x^*)},
\tag{18}$$

we have that

$$w_x(p_x^*) = \frac{F_x(p_x^*)}{\Pi_x''(p_x^*)} = \frac{b - p_x^*}{\Pi_x''(p_x^*)/f_x(p_x^*)}$$

Note that

$$\Pi_x''(p) = -2f_x(p) + f_x'(p)(b - p).$$

Therefore, also applying the first-order condition from equation (18), we have

$$\Pi_x''(p_x^*)/f_x(p_x^*) = -2 + \frac{F_x(p)f_x'(p)}{f_x^2(p)} = -2 + \sigma_x(p).$$

□

A similar argument yields Proposition 3.

**Proposition 3** (Sufficient revelation condition under DRC). *Under Assumptions 4 and 5, $V_{\mathrm{con}}(p^*) < V_{\mathrm{rev}}(p_0^*, p_1^*)$ if*

$$\frac{2 + L(p^*)\alpha_1(p^*)}{2 + L(p^*)\alpha_0(p^*)} > \frac{\theta F_1(p^*)/f_1(p^*)}{(1 - \theta)F_0(p^*)/f_0(p^*)},$$

*where $L(p) = \frac{b-p}{p}$ is the Lerner index, and $\alpha_x(p) = \frac{-pf_x'(p)}{f_x(p)}$ is the curvature of the task completion quantity function $F_x(p)$.*

PROOF. If $V_{\mathrm{const}}(r)$ is strictly monotone increasing in $r$, then $V_{\mathrm{con}}(p^*) < V_{\mathrm{rev}}(p_0^*, p_1^*)$. Since $V_{\mathrm{const}}(r)$ is strictly quasi-convex, a sufficient condition for $V_{\mathrm{const}}(r)$ to be strictly monotone increasing is $V_{\mathrm{const}}'(0) > 0$.

By equation (17), we have that $V_{\mathrm{const}}'(0) > 0$ if $\theta w_1(p^*) - (1 - \theta)w_0(p^*) > 0$.

$$w_x(p^*) = \frac{F_x(p^*)/f_x(p^*)}{-2 + (b - p^*)(f_x'(p^*)/f_x(p^*))} = \frac{F_x(p^*)/f_x(p^*)}{-2 - L(p^*)\alpha_x(p^*)}$$

Therefore, $\theta w_1(p^*) - (1 - \theta)w_0(p^*) > 0$ if

$$\frac{\theta F_1(p^*)/f_1(p^*)}{2 + L(p^*)\alpha_1(p^*)} < \frac{(1 - \theta)F_0(p^*)/f_0(p^*)}{2 + L(p^*)\alpha_0(p^*)}.$$

□

## F.3 Comparison of Decreasing Ratio Condition to Increasing Ratio Condition

The results in Section D.2.2 depend on the decreasing ratio condition (DRC) given in Assumption 5. This is analogous to the "increasing ratio condition (IRC)" from Aguirre et al. [4], which says that the ratio $\frac{W_x'(p)}{\Pi_x''(p)}$ is increasing in $p$. We now discuss in more detail the relationship between the DRC and the IRC, including sufficient conditions under which the DRC holds. In introducing the IRC, Aguirre et al. [4] describe a "very large set of demand functions" for which the IRC holds. Aguirre et al. [4] give sufficient conditions for the IRC to hold in Appendix B from their paper, which includes linear functions and exponential and constant elasticity functions.

For all of the sufficient conditions that Aguirre et al. [4] proposes for the IRC, these are also sufficient conditions for the DRC if paired with the additional condition that $\frac{F_x(p)}{f_x(p)}$ is increasing in $p$ for $x \in \{0, 1\}$. For example, a specific sufficient condition for the DRC, which also implies the IRC, is the following: Let $\sigma(p) = \frac{F(p)f'(p)}{f(p)^2}$. If $\sigma(p) \le 1$, and $\alpha(p) = -\frac{pf'(p)}{f(p)}$ is non-decreasing and positive in $p$, then the DRC holds. The IRC would also hold. A similar analogy can be made for all other conditions given in Appendix B of Aguirre et al. [4].

## F.4 Proofs from Section D.4

Lemmas 8 and 9 show that the principal always benefits from more information being revealed. Assumption 6 further implies that the principal strictly benefits from revelation.

**Lemma 8** (Principal prefers revelation). *Revealing $X$ never decreases the value of the principal: $\Pi_{\mathrm{rev}}(\rho^*) \ge \Pi_{\mathrm{con}}(p^*)$, where $\rho^* \in \arg\max_\rho \Pi_{\mathrm{rev}}(\rho)$. Revealing $X$ strictly increases the value of the principal only if $X$ and $C$ are not independent.*

PROOF. Assuming that the principal's feasible set of payments does not change between markets, the solution $\hat\rho(x) = p^*$ is in the feasible set of the principal's optimization problem with information revealed. Therefore,

$$\max_\rho \Pi_{\mathrm{rev}}(\rho) \ge \Pi_{\mathrm{rev}}(\hat\rho) = \Pi_{\mathrm{con}}(p^*).$$

If $X$ and $C$ are independent, we have $F_x = F$ for all $x \in X$, so

$$\max_\rho \Pi_{\mathrm{rev}}(\rho) = \max_\rho \mathbb{E}[F(\rho(X))(b - \rho(X))]$$

By Jensen's inequality, we have

$$\max_\rho \mathbb{E}[F(\rho(X))(b - \rho(X))] \le \mathbb{E}[\max_\rho F(\rho(X))(b - \rho(X))] = \mathbb{E}[F(p^*)(b - p^*)] = \Pi_{\mathrm{con}}(p^*).$$

Therefore, if $X$ and $C$ are independent, then $\Pi_{\text{rev}}(\rho^*) \leq \Pi_{\text{con}}(p^*)$. □

**Lemma 9** (Principal strictly benefits from revelation). Let $F_0$ and $F_1$ be continuously differentiable CDFs. If the MLRP holds (Assumption 6), then the principal strictly benefits when $X$ is revealed: $\Pi_{\text{rev}}(p_0^*, p_1^*) > \Pi_{\text{con}}(p^*)$.

PROOF. Since $F_0$, $F_1$ are continuously differentiable, the first-order necessary conditions hold for the optimal payments $p_0^*$, $p_1^*$ in equation (18). By these conditions, $p_0^* = p_1^*$ only if there exists a value $p$ such that

$$p + \frac{F_0(p)}{f_0(p)} = p + \frac{F_1(p)}{f_1(p)} = b.$$

Such a value $p$ cannot exist if $\frac{F_0(p)}{f_0(p)} \neq \frac{F_1(p)}{f_1(p)}$ for all $p$. The MLRP implies that $\frac{F_0(p)}{f_0(p)} < \frac{F_1(p)}{f_1(p)}$ for all $p$; therefore, $p_0^* \neq p_1^*$, and maximum value for $\Pi_{\text{rev}}$ is strictly greater than the maximum value for $\Pi_{\text{con}}$. □

## G    Additional Results on Information Revelation with Garbling

### G.1    Garbling and Prices

First, we analyze the effects of the information revelation amount $\varepsilon$ on the equilibrium prices set by the principal. Specifically, we show that both $p_0(\varepsilon), p_1(\varepsilon)$ are monotone with respect to $\varepsilon$.

**Lemma 11** (Monotonic price changes). Suppose $\theta = \frac{1}{2}$. Suppose the principal's utility is strictly concave as a function of price (Assumption 4). Suppose the MLRP holds (Assumption 6). Then $p_0'(\varepsilon) > 0$ and $p_1'(\varepsilon) < 0$ for all $\varepsilon \in [0, 1]$.

Lemma 11 acts as a sanity check that as the principal is aware of more information, the degree of differentiation between prices also increases. Furthermore, with less noise, the price in the higher-cost environment will only increase, and the price in the lower-cost environment will only decrease.

### G.2    Agent's Garbling Incentives: General Conditions

*G.2.1    Garbling condition under one zero-cost type.* As in Section D.2.1, we first analyze the restricted case where one of the agent types is anchored at zero-cost: suppose $C|X = 1$ takes value 0 with probability one. Proposition 4 gives a sufficient condition for the agent to prefer a non-zero amount of garbling over full revelation.

Let the conditional distribution of the cost $C$ given $Y$ be distributed with CDF $\mathbb{P}(C \leq c|Y = y) = G_y(c)$. When $\theta = \frac{1}{2}$,

$$G_0(c) = \frac{1+\varepsilon}{2}F_0(c) + \frac{1-\varepsilon}{2}F_1(c); \quad G_1(c) = \frac{1+\varepsilon}{2}F_1(c) + \frac{1-\varepsilon}{2}F_0(c).$$

Let $g_y(c)$ be denote the PDF.

**Proposition 4** (Sufficient garbling condition with zero-cost type). Suppose $\gamma = \theta = \frac{1}{2}$. Suppose $C|X = 1$ takes value 0 with probability 1. Suppose $F_0$ is continuously differentiable and $f_0(c)$ is bounded. $V_{\text{garb}}(\varepsilon)$ is maximized at $\varepsilon^* < 1$ if

$$\frac{b - p_0^*}{2 - \sigma_0(p_0^*)} < g_0(p_0^*), \tag{19}$$

where $g_0(p) = \int_0^p (1 - F_0(c))dc$ is the restricted mean cost of task completion, and $\sigma_0(p) = \frac{F_0(p)f_0'(p)}{f_0(p)^2}$ is the curvature of the inverse quantity function.

Notably, the inequality in equation (19) captures distributions that are not captured by Proposition 1. Thus, comparing Proposition 4 to Proposition 1 shows that the agent might want to garble, even if they may not always want to hide. In fact, the condition in equation (19) is quite general, and we show in the example in Section G.3 below that equation (19) applies to any log-concave Weibull distribution.

*G.2.2    General garbling condition.* Generalizing beyond the anchored setting, Proposition 5 gives a sufficient condition for the agent to prefer garbling over revelation for general $F_0, F_1$, which depends on similar identities. First, we generalize the restricted mean cost function $g_0(p)$ to a comparison of agent utilities.

**Definition 2** (Agent utility dominance). Let $\Delta(p_0, p_1) := (V_1(p_0) - V_1(p_1)) - (V_0(p_0) - V_0(p_1))$ denote the difference in sensitivities to the price change from $p_0$ to $p_1$ in each environment.

When $F_1$ exhibits first order stochastic dominance over $F_0$, we have that $\Delta(p_0, p_1) > 0$ for $p_0 > p_1$. The greater the dominance of $V_1(p)$ over $V_0(p)$ for all $p$, the greater the difference $\Delta$. Thus, we refer to $\Delta(p_0, p_1)$ as *agent utility dominance*. Using this definition, we now generalize the sufficient garbling condition from the anchored setting.

**Proposition 5** (Sufficient garbling condition). *Suppose $\gamma = \theta = \frac{1}{2}$. Suppose $F_0, F_1$ are continuously differentiable. $V_{\text{garb}}(\varepsilon)$ is maximized at $\varepsilon^* < 1$ if*

$$-\Pi_1'(p_0^*)\left(\frac{b - p_0^*}{2 - \sigma_0(p_0^*)}\right) - \Pi_0'(p_1^*)\left(\frac{b - p_1^*}{2 - \sigma_1(p_1^*)}\right) < \Delta(p_0^*, p_1^*), \tag{20}$$

*where $\sigma_x(p) = \frac{F_x(p) f_x'(p)}{f_x(p)^2}$ is the curvature of the inverse quantity function.*

The left hand side of the inequality in equation (20) is a weighted version of the difference $\frac{b - p_0^*}{2 - \sigma_0(p_0^*)} - \frac{b - p_1^*}{2 - \sigma_1(p_1^*)}$, which arises repeatedly in Aguirre et al. [4]'s analysis of the effects of price discrimination on total welfare, and also previously arose in Proposition 2. In cases where $\Pi_0'(p_1^*) > -\Pi_1'(p_0^*)$, the condition in Proposition 2 (equation (7)) would imply the condition in Proposition 5 (equation (20)), since $\Delta(p_0^*, p_1^*) \geq 0$.

## G.3 Example: Exponential and Weibull Distributions

We first illustrate the condition in Proposition 4 using an exponential distribution. Suppose $C|X = 1$ takes value 0 with probability one, and suppose $C|X = 0 \sim \text{Exp}(\frac{1}{\lambda_0})$, with $F_0$ defined as in equation (11). In this case, $g_0(p_0^*) = \lambda_0 F_0(p_0^*)$, and $\frac{b - p_0^*}{2 - \sigma_0(p_0^*)} = \lambda_0 F_0(p_0^*) \frac{1}{2 - F_0(p_0^*)}$. Therefore, the inequality in equation (19) holds for all $\lambda_0 > 0$.

The significance of this example is that if one agent type is anchored at 0, and the non-zero-cost environment induces an exponential distribution, the agent will *always* have an incentive to garble, regardless of the mean of the non-zero-cost distribution. Consider this in comparison to the exponential example from Section D.3, where the condition in Proposition 1 showed that agent prefers to fully conceal $X$ when $\lambda_0$ is small enough.

For a Weibull distribution with $F_0$ given by equation (13), we have for $k_0 \geq 1$,

$$g_0(p) = \frac{\lambda}{k_0}\left(\Gamma\left(\frac{1}{k_0}\right) - \Gamma\left(\frac{1}{k_0}, \frac{p^{k_0}}{\lambda_0^{k_0}}\right)\right).$$

If $k_0 \geq 1$, then the inequality in equation (19) from Proposition 4 holds for all $\lambda_0$. This encompasses all log-concave Weibull distributions. If $k_0 < 1$, then equation (19) does not necessarily hold, and fully flips for $k_0 < 0.5$.

Similarly to Figure 6, we can also consider simulations beyond the zero-cost anchored setting by considering all combinations of $\lambda_0, \lambda_1$ when $C|X = 0 \sim \text{Exp}(\frac{1}{\lambda_0})$ and $C|X = 1 \sim \text{Exp}(\frac{1}{\lambda_1})$, with $F_x$ given by equation (12). Figure 7 illustrates the combinations of $\lambda_0, \lambda_1$ for which the agent prefers some amount of garbling over full revelation. Figure 7 shows that $V_{\text{garb}}'(1) < 0$ as long one of the conditional means $\lambda_0$ or $\lambda_1$ is small enough. That is, as long as one of the revealed settings has low enough cost, the agent always prefers to garble, regardless of how high the cost of the other setting goes. This contrasts the revelation example in Section D.3, where even if $\lambda_1$ is close to 0, high enough $\lambda_0$ leads to the agent being willing to reveal.

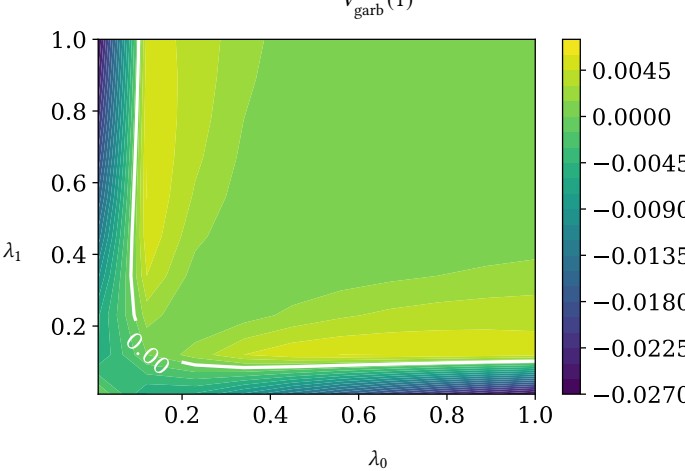

**Figure 7: Plot of $V_{\text{garb}}'(1)$ for a mixture of exponential distributions with means $\lambda_0, \lambda_1$. A negative value indicates that the agent prefers some amount of garbling over full revelation. As long as one of the revealed settings has low enough average cost, the agent always prefers some amount of garbling over full revelation, regardless of how high the mean is of the other setting.**

Finally, Figure 8 show a case in which the agent would prefer to reveal some amount of information via garbling over both concealment and revelation, but if *not* given the option to garble, then they would otherwise prefer concealment over revelation in the game from Figure 1.

## G.4 Principal's Garbling Preferences

Similarly to Section D.4, the principal always prefers for more information to be revealed. In fact, the garbling parameter $\varepsilon$ directly interpolates between the concealed and revealed utilities for the principal.

**Lemma 12.** $\Pi_{\text{con}}(p^*) \leq \Pi_{\text{garb}}(\varepsilon) \leq \Pi_{\text{rev}}(\rho^*)$ for all $\varepsilon$.

Also similarly to the comparison of full concealment and revelation settings, revealing less noisy information yields a strict improvement in the principal's utility if the MLRP holds.

**Lemma 13** (Strict principal improvement). Suppose $\theta = \frac{1}{2}$. Suppose the principal's utility is strictly concave (Assumption 4). If the MLRP holds (Assumption 6), then $\Pi'_{\text{garb}}(\varepsilon) > 0$ for all $\varepsilon \in [0, 1]$.

## G.5 Garbling vs. Restricted Price Discrimination

Our garbling model expands the agent's action space from a binary choice between full concealment and full revelation for a given $X$ to a continuous choice of revealing a garbled version parameterized by $\varepsilon$. Thus, $\varepsilon$ interpolates between the full concealment and full revelation settings.

Garbling is not the only way to interpolate between the full concealment and full revelation settings. In the price discrimination literature, there is an established model that interpolates between full price discrimination and no price discrimination by restricting that price difference between market segments can be no greater than some parameter $r$. Wright [52] models this restriction as arising from a "cost of transport" or arbitrage between two markets. Aguirre et al. [4] apply this interpolation by analyzing the marginal effect of $r$ on total welfare. We refer to this interpolation using $r$ as a *restricted price discrimination* model. We also leveraged this technique to analyze the agent's utility in Section D.2.2.

We now discuss in detail how the garbling model that we have introduced compares with this restricted price discrimination model. Specifically, we consider how the interpolation between concealment and revelation introduced through varying $\varepsilon$ in our garbling model compares to interpolation using a constraint parameter $r$.

In fact, the trajectory of the principal and agents' utilities as $r$ varies is different from the trajectory of the principal and agents' utilities as $\varepsilon$ varies. Most importantly to our setting, there is a qualitative difference between the functions $V_{\text{const}}(r)$ and $V_{\text{garb}}(\varepsilon)$. Lemma 7 shows that the value of $r$ that maximizes $V_{\text{const}}(r)$ always corresponds with either full concealment or full revelation. However, under the same conditions, the value of $\varepsilon$ that maximizes $V_{\text{garb}}(\varepsilon)$ is *not* always at the extremes, and is often somewhere in between 0 and 1. This is significant in our setting since the agent's power to choose $\varepsilon$ is directly built into the game, and the existence of an optimal $\varepsilon \in (0, 1)$ means that the agent benefits from the additional degree of freedom in their action space.

To visualize this difference between these interpolation methods, we can further map the combinations of principal and agent value onto the surplus triangle from Bergemann et al. [12]. Figure 8 shows an example where there exists an intermediate value $\varepsilon$ that the agent prefers over both concealment and revelation. In summary, both this example and Lemma 7 show that while there sometimes exist intermediate values $\varepsilon$ that the agent prefers over both concealment and revelation, this is notably *not* true for intermediate restrictions $r$ to the amount of price discrimination.

## H Proofs from Section G

Here we give proofs for results for the garbling model presented in Section G.

## H.1 Proofs from Section G.1

**Lemma 11** (Monotonic price changes). Suppose $\theta = \frac{1}{2}$. Suppose $F_0, F_1$ are continuously differentiable CDFs, the principal's utility is strictly concave (Assumption 4), and the MLRP holds (Assumption 6). Then $p'_0(\varepsilon) > 0$ and $p'_1(\varepsilon) < 0$ for all $\varepsilon \in [0, 1]$.

PROOF. $p_0(\varepsilon), p_1(\varepsilon)$ must satisfy first-order necessary conditions for optimality:

$$p_0(\varepsilon) + \frac{\frac{1+\varepsilon}{2}F_0(p_0(\varepsilon)) + \frac{1-\varepsilon}{2}F_1(p_0(\varepsilon))}{\frac{1+\varepsilon}{2}f_0(p_0(\varepsilon)) + \frac{1-\varepsilon}{2}f_1(p_0(\varepsilon))} = b; \quad p_1(\varepsilon) + \frac{\frac{1+\varepsilon}{2}F_1(p_1(\varepsilon)) + \frac{1-\varepsilon}{2}F_0(p_1(\varepsilon))}{\frac{1+\varepsilon}{2}f_1(p_1(\varepsilon)) + \frac{1-\varepsilon}{2}f_0(p_1(\varepsilon))} = b$$

Differentiating these first-order conditions, we have:

$$p'_0(\varepsilon) = \frac{\Pi'_0(p_0(\varepsilon)) - \Pi'_1(p_0(\varepsilon))}{-2\Pi''_{Y=0}(p_0(\varepsilon))}; \quad p'_1(\varepsilon) = \frac{\Pi'_1(p_1(\varepsilon)) - \Pi'_0(p_1(\varepsilon))}{-2\Pi''_{Y=1}(p_1(\varepsilon))}, \tag{21}$$

where

$$\Pi_{Y=0}(p) = \frac{1+\varepsilon}{2}\Pi_0(p) + \frac{1-\varepsilon}{2}\Pi_1(p); \quad \Pi_{Y=1}(p) = \frac{1+\varepsilon}{2}\Pi_1(p) + \frac{1-\varepsilon}{2}\Pi_0(p).$$

Figure 8: Trajectories of principal and agent utilities over $\varepsilon$ and $r$, mapped onto the triangle of possible combinations of principal and agent utilities from Bergemann et al. [12]. Here, cost is distributed as a mixture of exponentials with $C|X = 1 \sim \mathbf{Exp}(\frac{1}{\lambda_1})$, $C|X = 0 \sim \mathbf{Exp}(\frac{1}{\lambda_0})$, with $\lambda_0 = 0.5$, $\lambda_1 = 0.01$. The point $A$ corresponds to the concealed setting $(V_{\mathbf{con}}(p^*), \Pi_{\mathbf{con}}(p^*))$, and the point $F$ corresponds to the revealed setting $(V_{\mathbf{rev}}(p_0^*, p_1^*), \Pi_{\mathbf{rev}}(p_0^*, p_1^*))$. The solid blue line shows all combinations of $\Pi_{\mathbf{garb}}(\varepsilon), V_{\mathbf{garb}}(\varepsilon)$ for $\varepsilon \in [0, 1]$. The dashed orange line shows all combinations of $V_{\mathbf{const}}(r), \Pi_{\mathbf{const}}(r)$ for $r \in [0, p_0^* - p_1^*]$. First, note that the agent's utility at $A$ is higher than at $F$, so the agent prefers concealment over revelation for this particular $X$. However, there exists a point along the $\varepsilon$ trajectory in which $V_{\mathbf{garb}}(\varepsilon)$ achieves higher agent utility than the point $A$. However, this is *not* true of the $r$ trajectory. In general, Lemma 7 shows that intermediate values of $r$ will always be dominated by either the fully concealed or fully revealed settings.

Strict concavity from Assumption 4 makes both denominators of $p_x'(\varepsilon)$ positive.

The MLRP also implies that $p_0(\varepsilon) < p_0^*$ and $p_1(\varepsilon) > p_1^*$ for any $\varepsilon$. Therefore, by strict concavity of $\Pi_x(p)$, we have $\Pi_0'(p_0(\varepsilon)) > 0$, and $\Pi_1'(p_0(\varepsilon)) < 0$, implying that $p_0'(\varepsilon) > 0$. Similarly, $\Pi_1'(p_1(\varepsilon)) < 0$, and $\Pi_0'(p_1(\varepsilon)) > 0$, implying that $p_1'(\varepsilon) < 0$. □

## H.2 Proofs from Section 5.2

We first expand $V_{\mathrm{garb}}(\varepsilon)$ for $\theta = \frac{1}{2}$.

$$V_{\mathrm{garb}}(\varepsilon) = \frac{1}{2}\left(\frac{1+\varepsilon}{2}V_0(p_0(\varepsilon)) + \frac{1-\varepsilon}{2}V_1(p_0(\varepsilon)) + \frac{1+\varepsilon}{2}V_1(p_1(\varepsilon)) + \frac{1-\varepsilon}{2}V_0(p_1(\varepsilon))\right). \tag{22}$$

To prove Proposition 4, we first give Lemma 14 to handle the anchored zero-cost agent.

**Lemma 14.** Suppose $C|X = 1$ takes value 0 with probability 1. Suppose $f_0$ is bounded: $f_0(p) < B$ for all $p$ in the support. Then for any fixed value $b > 0$, there exists $\delta > 0$ such that for any $\varepsilon > \delta$, $p_1(\varepsilon) = 0$.

PROOF. Define $h(p, \varepsilon) = p + \frac{1}{f_0(p)}\frac{1+\varepsilon}{1-\varepsilon} + \frac{F_0(p)}{f_0(p)}$. By choosing $\delta$ close to 1, we can make the term $\frac{1+\delta}{1-\delta}$ arbitrarily large, and consequently $\frac{1}{f_0(p_1)}\frac{1+\delta}{1-\delta}$ arbitrarily large, since $f_0(p) > 0$. Then for any $b$, we choose $\delta$ close enough to 1 such that $\frac{1}{B}\frac{1+\delta}{1-\delta} > b$. □

**Proposition 4** (Sufficient garbling condition with zero-cost type). Suppose $\gamma = \theta = \frac{1}{2}$. Suppose $C|X = 1$ takes value 0 with probability 1. Suppose $F_0$ is continuously differentiable and $f_0(c)$ is bounded. $V_{\mathrm{garb}}(\varepsilon)$ is maximized at $\varepsilon^* < 1$ if

$$\frac{v - p_0^*}{2 - \sigma_0(p_0^*)} < g_0(p_0^*),$$

where $g_0(p) = \int_0^p (1 - F_0(c))dc$ is the restricted mean cost of task completion, and $\sigma_0(p) = \frac{F_0(p)f_0'(p)}{f_0(p)^2}$ is the curvature of the inverse quantity function.

PROOF. We show that the condition in equation (19) implies that $V_{\mathrm{garb}}'(1) < 0$.

For $C|X = 1$ taking value 0 with probability 1, we have $V_1(p) = p$. Substituting this into equation (22),

$$V_{\mathrm{garb}}(\varepsilon) = \frac{1}{2}\left(\frac{1+\varepsilon}{2}V_0(p_0(\varepsilon)) + \frac{1-\varepsilon}{2}p_0(\varepsilon) + \frac{1+\varepsilon}{2}p_1(\varepsilon) + \frac{1-\varepsilon}{2}V_0(p_1(\varepsilon))\right).$$

Differentiating this, we have

$$2V'_{\text{garb}}(\varepsilon) = p'_0(\varepsilon)\left(\frac{1+\varepsilon}{2}F_0(p_0(\varepsilon)) + \frac{1-\varepsilon}{2}\right) + p'_1(\varepsilon)\left(\frac{1+\varepsilon}{2} + \frac{1-\varepsilon}{2}F_0(p_1(\varepsilon))\right)$$

$$+ \frac{1}{2}\left((V_0(p_0(\varepsilon)) - V_0(p_1(\varepsilon))) - (p_0(\varepsilon) - p_1(\varepsilon))\right)$$

Evaluating this derivative at $\varepsilon = 1$, Lemma 14 implies that $p_1(\varepsilon) = 0$ and $p'_1(1) = 0$.

$$2V'_{\text{garb}}(1) = p'_0(1)F_0(p^*_0) + \frac{1}{2}(V_0(p^*_0) - p^*_0).$$

$$V_0(p^*_0) - p^*_0 = E[(p^*_0 - C)\,\mathbb{1}(C < p^*_0)|X = 0] - p^*_0 = -\frac{1}{2}g_0(p^*_0).$$

$$\implies 4V'_{\text{garb}}(1) = -g_0(p^*_0) + 2F_0(p^*_0)p'_0(1)$$

Simplifying $2F_0(p^*_0)p'_0(1)$:

$$2F_0(p^*_0)p'_0(1) = \frac{F_0(p^*_0)f_0(p^*_0)}{2f_0(p^*_0)^2 - F_0(p^*_0)f'_0(p^*_0)}$$

$$= \frac{\frac{F_0(p^*_0)}{f_0(p^*_0)}}{2 - \frac{F_0(p^*_0)f'_0(p^*_0)}{f_0(p^*_0)^2}}$$

$$= \frac{v - p^*_0}{2 - \sigma_0(p^*_0)}$$

Therefore,

$$V'_{\text{garb}}(1) < 0 \iff -g_0(p^*_0) + \frac{v - p^*_0}{2 - \sigma_0(p^*_0)} < 0.$$

$\square$

**Proposition 5** (Sufficient garbling condition). Suppose $\gamma = \theta = \frac{1}{2}$. Suppose $F_0, F_1$ are continuously differentiable. $V_{\text{garb}}(\varepsilon)$ is maximized at $\varepsilon^* < 1$ if

$$-\Pi'_1(p^*_0)\left(\frac{b - p^*_0}{2 - \sigma_0(p^*_0)}\right) - \Pi'_0(p^*_1)\left(\frac{b - p^*_1}{2 - \sigma_1(p^*_1)}\right) < \Delta(p^*_0, p^*_1).$$

Proof. Differentiating with respect to $\varepsilon$, we have

$$2V'_{\text{garb}}(\varepsilon) = p'_0(\varepsilon)\left(\frac{1+\varepsilon}{2}F_0(p_0(\varepsilon)) + \frac{1-\varepsilon}{2}F_1(p_0(\varepsilon))\right) + p'_1(\varepsilon)\left(\frac{1+\varepsilon}{2}F_1(p_1(\varepsilon)) + \frac{1-\varepsilon}{2}F_0(p_1(\varepsilon))\right)$$

$$+ \frac{1}{2}\left((V_0(p_0(\varepsilon)) - V_0(p_1(\varepsilon))) - (V_1(p_0(\varepsilon)) - V_1(p_1(\varepsilon)))\right).$$

Substituting in the price derivatives from equation (21) and the agent utility dominance identity from Definition 2,

$$2V'_{\text{garb}}(\varepsilon) = \left(\frac{\Pi'_0(p_0(\varepsilon)) - \Pi'_1(p_0(\varepsilon))}{-2\Pi''_{Y=0}(p_0(\varepsilon))}\right)\left(\frac{1+\varepsilon}{2}F_0(p_0(\varepsilon)) + \frac{1-\varepsilon}{2}F_1(p_0(\varepsilon))\right)$$

$$+ \left(\frac{\Pi'_1(p_1(\varepsilon)) - \Pi'_0(p_1(\varepsilon))}{-2\Pi''_{Y=1}(p_1(\varepsilon))}\right)\left(\frac{1+\varepsilon}{2}F_1(p_1(\varepsilon)) + \frac{1-\varepsilon}{2}F_0(p_1(\varepsilon))\right)$$

$$+ \frac{1}{2}\left(-\Delta(p_0(\varepsilon), p_1(\varepsilon))\right).$$

Let

$$z^\varepsilon_0(p) = \frac{V'_{Y=0}(p)}{\Pi''_{Y=0}(p)} = \frac{\frac{1+\varepsilon}{2}F_0(p) + \frac{1-\varepsilon}{2}F_1(p)}{\Pi''_{Y=0}(p)},$$

$$z^\varepsilon_1(p) = \frac{V'_{Y=1}(p)}{\Pi''_{Y=1}(p)} = \frac{\frac{1+\varepsilon}{2}F_1(p) + \frac{1-\varepsilon}{2}F_0(p)}{\Pi''_{Y=1}(p)}.$$

Then

$$4V'_{\text{garb}}(\varepsilon) = \left(\Pi'_0(p_0(\varepsilon)) - \Pi'_1(p_0(\varepsilon))\right)\left(-z^\varepsilon_0(p_0(\varepsilon))\right) + \left(\Pi'_1(p_1(\varepsilon)) - \Pi'_0(p_1(\varepsilon))\right)\left(-z^\varepsilon_1(p_1(\varepsilon))\right)$$
$$- \Delta(p_0(\varepsilon), p_1(\varepsilon)).$$

Evaluating this at $\varepsilon = 1$, we have

$$4V'_{\text{garb}}(1) = \left(\Pi'_0(p^*_0) - \Pi'_1(p^*_0)\right)\left(-z_0(p^*_0)\right) + \left(\Pi'_1(p^*_1) - \Pi'_0(p^*_1)\right)\left(-z_1(p^*_1)\right)$$
$$- \Delta(p^*_0, p^*_1),$$

where

$$z_x(p) = \frac{V'_x(p)}{\Pi''_x(p)}.$$

Note that $V'_x(p^*_x) = W'_x(p^*_x)$, so at $p^*$, this is identical to the $z$ function from Aguirre et al. [4]. By first-order optimality conditions,

$$-z_x(p^*_x) = \frac{b - p^*_x}{2 - \sigma_x(p^*_x)}.$$

Therefore, $V'_{\text{garb}}(1) < 0$ if

$$\left(-\Pi'_1(p^*_0)\right)\left(\frac{b - p^*_0}{2 - \sigma_0(p^*_0)}\right) + \left(-\Pi'_0(p^*_1)\right)\left(\frac{b - p^*_1}{2 - \sigma_1(p^*_1)}\right) < \Delta(p^*_0, p^*_1).$$

$\square$

## H.3 Proofs from Section G.4

**Lemma 12.** $\Pi_{\text{con}}(p^*) \leq \Pi_{\text{garb}}(p_0(\varepsilon), p_1(\varepsilon)) \leq \Pi_{\text{rev}}(p^*_0, p^*_1)$ for all $\varepsilon$.

PROOF. For $\varepsilon \in \{0, 1\}$, the inequalities clearly hold. Fix $\varepsilon \in (0, 1)$. For the lower bound,

$$\Pi_{\text{garb}}(p_0(\varepsilon), p_1(\varepsilon)) \geq \Pi_{\text{garb}}(p^*, p^*) = \Pi_{\text{con}}(p^*).$$

For the upper bound, since the noise $\xi$ is independent of $X$ and $C$,

$$\Pi_{\text{garb}}(p_0, p_1) = \phi(\theta, \gamma)\Pi_{\text{rev}}(p_0, p_1) + \psi(\theta, \gamma)\Pi_{\text{rev}}(p_1, p_0) \leq \Pi_{\text{rev}}(p^*_0, p^*_1)$$

where $\phi, \psi$ are some positive functions of $\theta, \gamma$. $\square$

**Lemma 13** (Strict principal improvement). Suppose $\gamma = \theta = \frac{1}{2}$. Suppose the principal's utility is strictly concave (Assumption 4). If the MLRP holds (Assumption 6), then $\Pi'_{\text{garb}}(\varepsilon) > 0$ for all $\varepsilon \in [0, 1]$.

PROOF.

$$\Pi_{\text{garb}}(\varepsilon) = \frac{1}{2}(b - p_0(\varepsilon))\left(\frac{1 + \varepsilon}{2}F_0(p_0(\varepsilon)) + \frac{1 - \varepsilon}{2}F_1(p_0(\varepsilon))\right)$$
$$+ \frac{1}{2}(b - p_1(\varepsilon))\left(\frac{1 + \varepsilon}{2}F_1(p_1(\varepsilon)) + \frac{1 - \varepsilon}{2}F_0(p_1(\varepsilon))\right).$$

Differentiating with respect to $\varepsilon$:

$$4\Pi'_{\text{garb}}(\varepsilon) = F_0(p_0(\varepsilon))(b - p_0(\varepsilon)) - F_0(p_1(\varepsilon))(b - p_1(\varepsilon))$$
$$+ F_1(p_1(\varepsilon))(b - p_1(\varepsilon)) - F_1(p_0(\varepsilon))(b - p_0(\varepsilon))$$

By the MLRP and strict concavity of $\Pi_0(p), \Pi_1(p)$, we have that $p_1(\varepsilon) < p_0(\varepsilon) < p^*_0$. Strict concavity of $\Pi_0(p)$ and the optimality of $p^*_0$ for $\Pi_1$ then implies that

$$F_0(p_1(\varepsilon))(v - p_1(\varepsilon)) < F_0(p_0(\varepsilon))(v - p_0(\varepsilon)).$$

Similarly, by the MLRP and strict concavity of $\Pi_0(p), \Pi_1(p)$, we have that $p^*_1 < p_1(\varepsilon) < p_0(\varepsilon)$. Strict concavity of $\Pi_1(p)$ and the optimality of $p^*_1$ for $\Pi_1$ implies that

$$F_1(p_0(\varepsilon))(v - p_0(\varepsilon)) < F_1(p_1(\varepsilon))(v - p_1(\varepsilon)).$$

$\square$

# I Further Uber and Lyft Experiment Details

Here we provide additional details on the experiment setup and implementation using the Uber and Lyft dataset.

## I.1 Scenario 2: Revelation to Refine an Existing Pricing Model.

As an additional scenario, we suppose that the principal starts with a slightly more sophisticated pricing model as a baseline. Suppose Uber decides to offer a non-negative price adjustment $p$ on top of their existing pricing model as an incentive for drivers to switch to their platform. The agent can choose to reveal $X$ = Lyft's surge multiplier (which Uber is otherwise unable to observe), in which case Uber's price adjustment would depend on $X$. To approximate Uber's initial pricing model, we train a linear model targeting Uber's price over all features in Uber's dataset.[5] Let $C$ be the difference between Uber's estimated price (on the Lyft dataset) and Lyft's price.

*I.1.1    Results.* We present results with the same switching value $b$ as earlier. Figure 9 shows that the agent also prefers to reveal $X$ = Lyft's surge multipler. Letting $Z^t$ be a binarized version of the surge multiplier, Figure 9 also shows that there exist at least one value of $t$ for which the agent prefers garbling over revelation. Since the surge multiplier is never less than 1 in the data, and the agent already prefers to reveal for $t = 1$, we do not observe any thresholds for which the agent strictly prefers to conceal. However, the agent's value for revelation still decays to 0 as the threshold increases to its maximum. In this case, the agent does not gain any value from additional garbling on top of revelation. Overall, this scenario showed that a cost-correlated feature like the surge multiplier was beneficial to an agent to reveal, even when the principal platform's existing pricing model already depended on other features like distance.

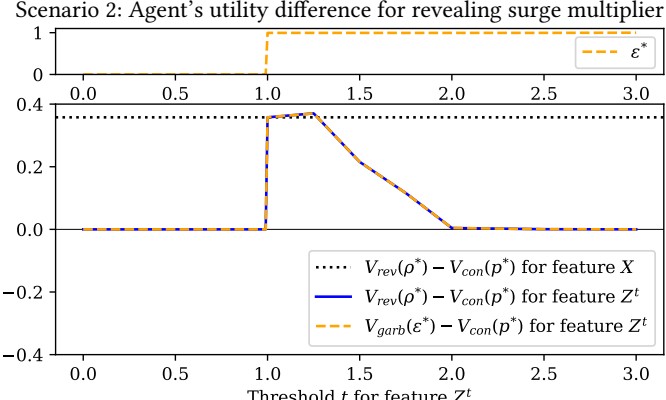

Figure 9: Differences between the agent's revealed and concealed utilities in Scenario 2, with $X$ = Lyft's surge multiplier. The labels are the same as Figure 4.

## I.2    Estimating the Principal's and Agent's Utilities

*For binarized features:* For a binarized feature $Z^t$, we directly estimate the conditional pdfs $f_0$ and $f_1$ using kernel density estimation over the data conditioned on $Z^t = 0, 1$ respectively. Specifically, we use the Scipy `gaussian_kde` method with default bandwidth [49]. The principal and agents' revealed values are built directly on $f_0$, $f_1$, and the empirical estimate for $\theta$. The hidden cost distribution is given by $f(c) = (1 - \theta)f_0(c) + \theta * f_1(c)$.

*For full features:* When estimating the agent's value difference for revealing a full non-binarized feature $X$, we substitute the empirical expectation over the dataset for both the principal and the agent's values. That is, for a dataset with points $\{(C_i, X_i)\}_{i=1}^{n}$, we estimate the principal's revealed value as

$$\Pi_{\text{rev}}(\rho) = \mathbb{E}[\mathbb{1}(C < \rho(X))(b - \rho(X))] \approx \frac{1}{n}\sum_{i=1}^{n}\mathbb{1}(C_i < \rho(X_i))(b - \rho(X_i)).$$

Similarly, we estimate the agent's revealed value as

$$V_{\text{rev}}(\rho) = \mathbb{E}[\mathbb{1}(C < \rho(X))(\rho(X) - C)] \approx \frac{1}{n}\sum_{i=1}^{n}\mathbb{1}(C_i < \rho(X_i))(\rho(X_i) - C_i).$$

The principal and agents' hidden values are also estimated empirically using only the points $\{C_i\}_{i=1}^{n}$.

These estimates are then used to compute the optimal hidden price $p^*$, revealed price function $\rho^*$, and the agent's value differences.

---

[5]A linear model is, of course, still far from Uber's actual pricing methodology, but the purpose of this illustration is to have a model that depends on more features initially.

## I.3    Parameterizing $\rho(x)$ for Continuous $X$

When the variable $X$ revealed is continuous, we parameterize the principal's pricing function $\rho(x)$ as linear for simplicity and tractability. This is more restrictive than any non-parametric and arbitrarily expressive $\rho(x)$, though not uncommon in practice. In the experiments, the agent also operates under the assumption that $\rho(x)$ would be linear. For the purposes of our model, the most important assumption is that the agent is aware of the family of the pricing functions over which the principal is optimizing, $\rho \in \mathcal{F}$. The value difference for the agent between revealing and concealing will ultimately also depend on the family $\mathcal{F}$.

## I.4    Garbling Amounts

For each $Z^t$, we compute the optimal garbling amount

$$\varepsilon^* = \arg\max_{\varepsilon \in [0,1]} \; V_{\text{garb}}(\varepsilon).$$

Notably, for the garbling distribution $\xi$, we set $\gamma = \theta = P(Z^t = 1)$ for each $Z^t$. This ensures that the marginal distribution of the garbled variable $Y$ matches the marginal distribution of $Z^t$ for each $t$.

## I.5    Scenario 2 Costs

To construct the cost variable $C$ for Scenario 2, we first estimate Uber's pricing model by training an ordinary least squares linear regression model on Uber's dataset using the price column as the target, and all features other than the surge multiplier as inputs. The $R^2$ score on the training data is 0.1131. We use Scikit Learn's `linear_model` function [44].

The column representing the cost $C$ is then computed on the Lyft dataset as $\max(0, \text{Uber's predicted price} - \text{Lyft's price})$. This produces a non-negative cost variable.

Note that the principal optimizing its price using this non-negative cost variable is equivalent to the principal solving the constrained optimization problem for price restricted to non-negative prices only.

## I.6    Results for Different Values of $b$

Figures 4 and 9 show results when the principal's value is $b = 150$, or $1.5\bar{C}$, where $\bar{C} = 100$. Figures 10 and 11 show the same analysis for $b \in \{50, 100, 200\}$.

Received 20 February 2007; revised 12 March 2009; accepted 5 June 2009

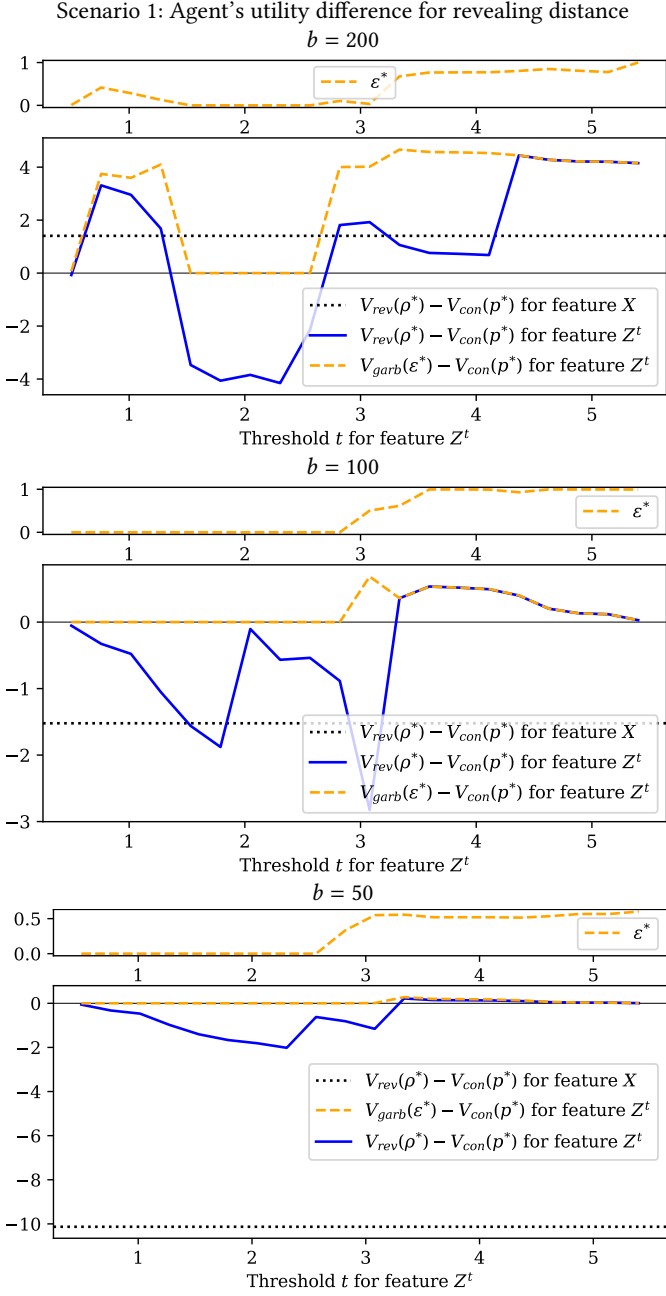

**Figure 10: Differences between the agent's revealed and concealed utilities in the Scenario from Section 6, with $X$ = distance, for different values of $b$. The labels are the same as Figure 4.**

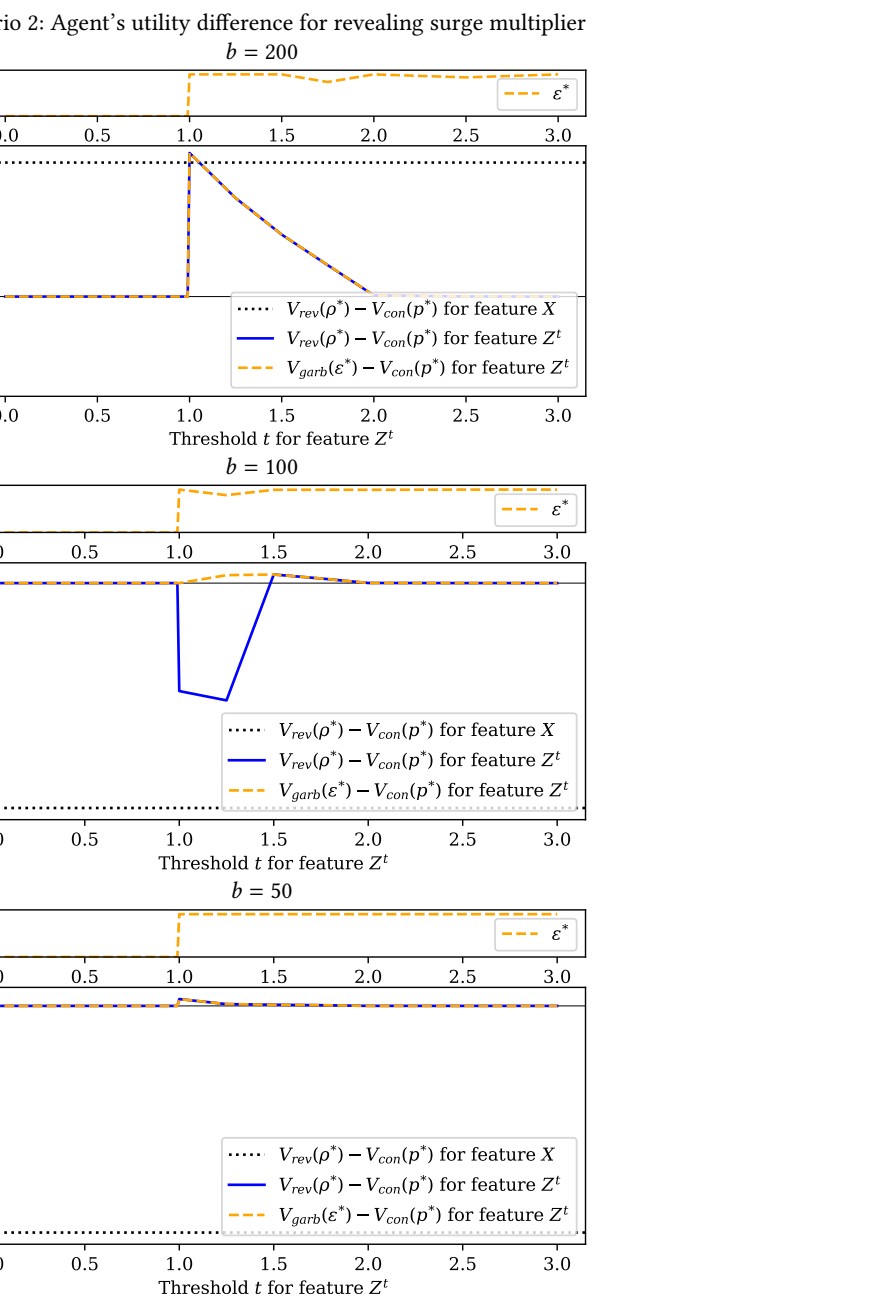

**Figure 11: Differences between the agent's revealed and concealed utilities in Scenario 2, with $X$ = Lyft's surge multiplier, for different values of $b$. The labels are the same as Figure 4.**

