# OpenReview forum: "Relying on the Metrics of Evaluated Agents"
_ACM.org/TheWebConf/2025/Conference — WWW 2025 Oral_

### Official Review · Reviewer_paTJ · 2024-11-19

**Novelty:** 5
**Technical Quality:** 4

**Review:**

The paper proposes a principal-agent model to study the problem of eliciting evaluation metrics (variables that are correlated with the agent's performance) from agents directly. Under several modeling assumptions, the paper studies when will the agents be willing to reveal such a metric that is unknown to the principal, and when will more information benefit the social welfare of the system. Theoretical insights are further verified using real-world Uber-Lyft data.

Pros:
* For the proposed problem, the paper considers both a binary metric model and the continuous garbling model. The binary model is simple and insightful while the garbling model is more general and flexible. Similar insights are gained under these models which increases the robustness of the paper.
* The paper is overall well-written. Key assumptions and results are clearly illustrated and explained. Intuitions of the theorems are provided.


Cons:
* The application of the study seems very limited. The results are useful and novel only when the agent knows a metric that the principal does not know its existence. If the principal has some prior of the metric, the problem reduces to a classic principal-agent problem. However, it's not clear where such a setting is sound and important. For the experiments on Uber data, as said in footnote 3, the metric of distance is usually known by the platform (the principal). Furthermore, the distance seems to be a variable that is better known than the cost in this scenario.
* Some of the model assumptions are very strong. For example, assumptions 1 and 2 put strong requirements on the cost distribution which seem unlikely to hold for general settings. Although the insights derived from this model are partially verified using real data, less is known whether these results are insightful for a cost distribution that strongly violates these assumptions.
* Several results are speaking with the social welfare of the system. However, it seems whether the revealed information benefits the principal is a more important question to study. That is, even though revealing the metric is beneficial for the welfare if it harms the principal, such a function will not be implemented because the principal is the decision maker.

**Questions:**

Please refer to my review.

In addition, could the author specify which assumptions are violated in the experiments using the real data? Why are the experiments necessary given that results are proven in theorems?

**Reviewer Confidence:**

3: The reviewer is confident but not certain that the evaluation is correct

**Scope:**

3: The work is somewhat relevant to the Web and to the track, and is of narrow interest to a sub-community

---

### Official Review · Reviewer_1opm · 2024-11-27

**Novelty:** 5
**Technical Quality:** 4

**Review:**

**Summary**

This paper considers a contracting problem between a principal and an agent. The model follows the classical contract design with Bayesian cost of effort. The novelty is that the agent may hold a private signal unknown to the principal. Particularly, the signal is a binary with a threshold on the cost of effort. The contributions include: 1) conditions where the agent selects to reveal or hide the signal; 2) the welfare change with an unknown signal; 3) revealing noisy signals increases welfare; and 4) experiments on ridesharing data.


**Strengths**

 * The paper is well-written and easy to follow.
* The welfare analysis in the garbling section is interesting. I wonder if there is a characterization of the optimal garbling.


**Weakness**

* I'm not fully convinced by the model. The beginning of the paper seems to suggest a one-shot game where designing a metric is costly. However, the experiment is ridesharing, a setting with repeated interactions, where the platform is fully aware of the hidden information.  But if talking in a single-shot game, or a game with fewer rounds of interactions, why does the paper assume truthful reporting of the information structure, which can hardly be verified? I found lines 165-168 in the introduction particularly confusing, which seems to say the platform has data collection power yet knows nothing about the hidden signal.

* Some model choices are not fully justified. E.g. Do results generalize to non-threshold signals? In the setting where agents are allowed to garble, why does the paper restrict attention to binary signals? Is this without loss or does it explain real applications better?

**Minor comments**

* line 203, the model for "elicitation" of unknown metrics: I'm not sure if this is elicitation since the principal does not know if she's eliciting information or what is being elicited. I'd view it as an analysis of strategic revealing behavior.
* line 434-435, typo: "low cost type" appearing twice.

**Questions:**

See my comments above about model choices.

**Reviewer Confidence:**

3: The reviewer is confident but not certain that the evaluation is correct

**Scope:**

4: The work is relevant to the Web and to the track, and is of broad interest to the community

---

### Official Review · Reviewer_MjmS · 2024-12-02

**Novelty:** 6
**Technical Quality:** 6

**Review:**

Summary
The authors present a model to analyze the discovery of metrics in online platforms. They consider metrics that an agency knows but the evaluators do not and analyze the incentives that influence the agent to share the metric with an evaluator. They model the interaction between agents and evaluators as an agency game and found that an agent will prefer to reveal metrics that highlight the most difficult tasks and conceal metrics that highlight the easiest tasks. The authors also introduce the option of garbling a metric as an alternative to fully revealing or hiding it. They show that garbling yields Pareto improvement for both the agent and evaluator. They then demonstrate how their theory can apply to real scenarios and feature distributions using public data from ride-share platforms Uber and Lyft.

The paper is clear, well-written and structured. It introduces the problem, explains the proposed solution, and highlight their contributions. While I have little agency theory and economics background, their proposal is intuitive and technically sound with rigorous mathematical modeling and analysis. Furthermore, their findings have important implications for the design of evaluation metrics in online platforms and regulatory settings.

Pros
* The model addresses an important problem.
* The analysis is rigorous and the assumptions are well-justified.
* The use of real-world ride-share data shows the model's potential applicability to practical scenarios.
* The introduction of garbling mechanisms to address privacy concerns aligns with ethical data handling in online platforms.

Cons (For improvement/future work)
* The model focuses on a single agent but there are also cases in which multiple agents interact with the evaluator.
* The paper primarily considers economic incentives but other factors like reputation or fairness may also influence information sharing.

**Questions:**

* The model focuses on a single agent, but there are also cases in which multiple agents interact with the evaluator. How can the model be extended to analyze such multi-agent settings?

* How does the choice of garbling mechanism affect the trade-off between privacy and the economic benefits of information sharing?

* What other real-world data could this be applicable to?

**Reviewer Confidence:**

3: The reviewer is confident but not certain that the evaluation is correct

**Scope:**

4: The work is relevant to the Web and to the track, and is of broad interest to the community

---

### Official Review · Reviewer_HVd3 · 2024-12-02

**Novelty:** 6
**Technical Quality:** 6

**Review:**

This paper models the incentives of an evaluated agent (e.g., a rideshare driver) to reveal or conceal metrics to an evaluator (e.g., a rideshare platform). The authors show that agents are more likely to reveal metrics highlighting high-cost tasks and conceal those indicating low-cost ones. Interestingly, adding noise to a metric can benefit both the agent and the evaluator, creating an economic value for privacy. This is demonstrated through a theoretical model and an empirical analysis using rideshare data. The findings offer insights into information elicitation and the design of effective evaluation metrics in online platforms.


Overall, I really liked this paper and found it quite well-written. I don't have a lot of meaningful comments on the execution per se, I find it well above an acceptable standard.

*One suggestion would be to connect this to the work of Ederer, Holden and Meyer (Rand 201x).

*There's also the literature on unawareness in the literature on epistemic game theory (for an application to contract theory, see e.g. Emel Filiz Ozbay's work) which is highly related to the idea of unknown unknowns (though frankly I like this paper's simple modeling of it more than that body of work).

* Finally, this appears to be a persuasion / privacy version of the literature on strategic disclosure (starting from Dye 1985). That paper has been incredibly influential in both accounting, and strategic communication within econ. This leads me to wonder whether there's an interesting follow-up thinking about the connections between using privacy tech to improve strategic disclosure in accounting settings (e.g., we verifiably add noise to e.g. earnings reports to improve quality of information transmitted). I realize this may not be up the authors' research alley, but similar ideas have been proposed in different settings, and improving the efficiency of capital markets may be incredibly valuable.

**Questions:**

I've stated my main suggestions above, no further questions.

**Reviewer Confidence:**

3: The reviewer is confident but not certain that the evaluation is correct

**Scope:**

4: The work is relevant to the Web and to the track, and is of broad interest to the community